# Proving the Limited Scalability of Centralized Distributed Optimization via a New Lower Bound Construction

**Alexander Tyurin**
AXXX, Moscow, Russia
Applied AI Institute, Moscow, Russia
alexandertiurin@gmail.com

## Abstract

We consider centralized distributed optimization in the classical federated learning setup, where $n$ workers jointly find an $\varepsilon$–stationary point of an $L$–smooth, $d$–dimensional nonconvex function $f$, having access only to unbiased stochastic gradients with variance $\sigma^2$. Each worker requires at most $h$ seconds to compute a stochastic gradient, and the communication times from the server to the workers and from the workers to the server are $\tau_{\mathrm{s}}$ and $\tau_{\mathrm{w}}$ seconds per coordinate, respectively. One of the main motivations for distributed optimization is to achieve scalability with respect to $n$. For instance, it is well known that the distributed version of SGD has a variance-dependent runtime term $h\sigma^2 L\Delta/n\varepsilon^2$, which improves with the number of workers $n$, where $\Delta := f(x^0) - f^*$, and $x^0 \in \mathbb{R}^d$ is the starting point. Similarly, using unbiased sparsification compressors, it is possible to reduce *both* the variance-dependent runtime term and the communication runtime term from $\tau_{\mathrm{w}} dL\Delta/\varepsilon$ to $\tau_{\mathrm{w}} dL\Delta/n\varepsilon + \sqrt{\tau_{\mathrm{w}} dh\sigma^2/n\varepsilon} \cdot L\Delta/\varepsilon$, which also benefits from increasing $n$. However, once we account for the communication from the server to the workers $\tau_{\mathrm{s}}$, we prove that it becomes infeasible to design a method using unbiased random sparsification compressors that scales both the server-side communication runtime term $\tau_{\mathrm{s}} dL\Delta/\varepsilon$ and the variance-dependent runtime term $h\sigma^2 L\Delta/\varepsilon^2$, better than poly-logarithmically in $n$, even in the homogeneous (i.i.d.) case, where all workers access the same function or distribution. Indeed, when $\tau_{\mathrm{s}} \simeq \tau_{\mathrm{w}}$, our lower bound is $\tilde{\Omega}\left(\min\left\{h\left(\frac{\sigma^2}{n\varepsilon}+1\right)\frac{L\Delta}{\varepsilon} + \tau_{\mathrm{s}} d\frac{L\Delta}{\varepsilon},\; h\frac{L\Delta}{\varepsilon} + h\frac{\sigma^2 L\Delta}{\varepsilon^2}\right\}\right)$. To establish this result, we construct a new "worst-case" function and develop a new lower bound framework that reduces the analysis to the concentration of a random sum, for which we prove a concentration bound. These results reveal fundamental limitations in scaling distributed optimization, even under the homogeneous (i.i.d.) assumption.

## 1 Introduction

We focus on the classical federated learning setup, where $n$ workers, such as CPUs, GPUs, servers, or mobile devices, are connected to a central server via a communication channel (Konečný et al., 2016; McMahan et al., 2017). All workers collaboratively solve a common optimization problem in a distributed fashion by computing stochastic gradients and sharing this information with the server, which then propagates it to other workers. Together, they aim to minimize a smooth nonconvex objective function defined as

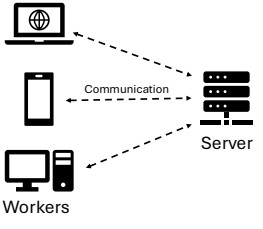

Communication

Server

Workers

$$\min_{x \in \mathbb{R}^d} f(x), \tag{1}$$

where $f : \mathbb{R}^d \to \mathbb{R}$ and $d$ is the dimension of $f$. We assume that $d$ is huge, which is indeed the case in modern machine learning and large language model training (Brown et al., 2020; Touvron et al., 2023).

We consider the *homogeneous* (i.i.d.) setting, where all workers have access to stochastic gradients of the same underlying function $f$. **As the reader will see, the homogeneous setting assumption is a challenge, not a limitation of our work**: all results extend, potentially with even stronger implications, to the more general *heterogeneous* (non-i.i.d.) case, when each worker $i$ works with $f_i \neq f$. We consider the standard assumptions:

**Assumption 1.1.** $f$ is differentiable & $L$–smooth, i.e., $\|\nabla f(x) - \nabla f(y)\| \leq L\|x - y\|, \forall x, y \in \mathbb{R}^d$.

**Assumption 1.2.** There exist $f^* \in \mathbb{R}$ such that $f(x) \geq f^*$ for all $x \in \mathbb{R}^d$. We define $\Delta := f(x^0) - f^*$, where $x^0$ is a starting point of methods.

For all $i \in [n]$, worker $i$ calculates unbiased stochastic gradients $\nabla f(x; \xi)$ with $\sigma^2$-variance-bounded variances, where $\xi$ is a random variable with some distribution $\mathcal{D}_\xi$.

**Assumption 1.3** (Homogeneous setting). For all $i \in [n]$, worker $i$ can only calculate $\nabla f(x; \xi)$ and $\mathbb{E}_\xi[\nabla f(x; \xi)] = \nabla f(x)$ and $\mathbb{E}_\xi[\|\nabla f(x; \xi) - \nabla f(x)\|^2] \leq \sigma^2$ for all $x \in \mathbb{R}^d$, where $\sigma^2 \geq 0$.

The goal in the nonconvex world is to find an $\varepsilon$–stationary point, a (random) point $\bar{x} \in \mathbb{R}^d$ such that $\mathbb{E}[\|\nabla f(\bar{x})\|^2] \leq \varepsilon$ (Nemirovskij & Yudin, 1983; Murty & Kabadi, 1985). We also consider a realistic computation and communication scenario:

**Assumption 1.4.** Each of the $n$ workers requires at most $h$ seconds to compute a stochastic gradient, and communication *from the server to any worker* (s2w communication) takes at most $\tau_s$ seconds per coordinate, and communication *from any worker to the server* (w2s communication) takes at most $\tau_w$ seconds per coordinate.

For instance, under Assumption 1.4, it takes $d \times \tau_s$ and $d \times \tau_w$ seconds to send a vector $v \in \mathbb{R}^d$ from the server to any worker and from any worker to the server, respectively. We consider settings with bidirectional communication costs, where communication in both directions requires time. Typically, especially in the early stages of federated learning algorithm development, most works assume that communication *from the server to the workers* is free, i.e., $\tau_s = 0$, which is arguably not true in practice: communication over the Internet or 4G/5G networks can be costly in both directions (Huang et al., 2012; Narayanan et al., 2021).

## 1.1 RELATED WORK

**1. Communication is free.** Let us temporarily assume that communication does not take time, i.e., $\tau_s = 0$ and $\tau_w = 0$. Then, in this scenario, the theoretically fastest strategy is to run the Synchronized SGD method, i.e., $x^{k+1} = x^k - \frac{\gamma}{n}\sum_{i=1}^{n}\nabla f(x^k; \xi_i^k)$, where $\gamma = \Theta(\min\{1/L, \varepsilon n/L\sigma^2\})$, $\{\xi_i^k\}$ are i.i.d., and $\{\nabla f(x^k; \xi_i^k)\}$ are computed in parallel by the workers, which send to the server that aggregates and calculates $x^{k+1}$. One can show that the time complexity of this method is $\mathcal{O}\left(h\left(\frac{L\Delta}{\varepsilon} + \frac{\sigma^2 L\Delta}{n\varepsilon^2}\right)\right)$, because it requires $\mathcal{O}\left(\frac{L\Delta}{\varepsilon} + \frac{\sigma^2 L\Delta}{n\varepsilon^2}\right)$ iterations (Lan, 2020), and each iteration takes at most $h$ seconds due to Assumption 1.4. Moreover, this result is *optimal and can not be improved* (Arjevani et al., 2022; Tyurin & Richtárik, 2023b).

*Observation 1:* One obvious but important observation is that the second "statistical term" in the complexity bound scales with $n$. The larger the number of workers $n$, the smaller the overall time complexity of Synchronized SGD, with a linear improvement in $n$. This is a theoretical justification for the importance of distributed optimization and the use of many workers.

**2. Worker-to-server communication can not be ignored.** For now, consider the setup where communication from workers to the server takes $\tau_w > 0$ seconds, while communication from the server to the workers is free, i.e., $\tau_s = 0$. In this scenario, the described version of Synchronized SGD has a suboptimal $\mathcal{O}\left(h\left(\frac{L\Delta}{\varepsilon} + \frac{\sigma^2 L\Delta}{n\varepsilon^2}\right) + \tau_w d\left(\frac{L\Delta}{\varepsilon} + \frac{\sigma^2 L\Delta}{n\varepsilon^2}\right)\right)$ time complexity, because it takes $h$ seconds to calculate a stochastic gradient and $\tau_w d$ seconds to send the stochastic gradients of size $d$ to the server, which calculates $x^{k+1}$. However, if we slightly modify this method and consider Batch Synchronized SGD:

$$x^{k+1} = x^k - \frac{\gamma}{n}\sum_{i=1}^{n}\frac{1}{b}\sum_{j=1}^{b}\nabla f(x^k; \xi_{ij}^k) \qquad \text{(Batch Synchronized SGD)}$$

with $b = \Theta(\sigma^2/\varepsilon n)$ and $\gamma = \Theta(1/L)$, then the time complexity becomes

$$\mathcal{O}\left(h\left(\frac{L\Delta}{\varepsilon} + \frac{\sigma^2 L\Delta}{n\varepsilon^2}\right) + \tau_w d\frac{L\Delta}{\varepsilon}\right), \tag{2}$$

because the number of iterations reduces to $\mathcal{O}\left(L\Delta/\varepsilon\right)$. In other words, each worker, instead of immediately sending a gradient, locally aggregates a batch of size $b$ to reduce the number of communications. It turns out that the last complexity can be improved further with the help of unbiased compressors:

**Definition 1.5.** A mapping $\mathcal{C} : \mathbb{R}^d \times \mathbb{S}_\nu \to \mathbb{R}^d$ with a distribution $\mathcal{D}_\nu$ is an *unbiased compressor* if there exists $\omega \geq 0$ such that $\mathbb{E}_\nu[\mathcal{C}(x;\nu)] = x$ and $\mathbb{E}_\nu[\|\mathcal{C}(x;\nu) - x\|^2] \leq \omega\|x\|^2$ for all $x \in \mathbb{R}^d$. We $\mathbb{U}(\omega)$ denote the family of such compressors. The community uses the shorthand $\mathcal{C}(x;\nu) \equiv \mathcal{C}(x)$, which we also follow.

A standard example of an unbiased compressor is $\text{Rand}K \in \mathbb{U}(d/K - 1)$, which selects $K$ random coordinates of the input vector $x$, scales them by $d/K$, and sets the remaining coordinates to zero (see Def. C.1). Numerous other examples of unbiased compressors have been explored in the literature (Beznosikov et al., 2020; Xu et al., 2021; Horváth et al., 2022; Szlendak et al., 2021). Using the seminal ideas (Seide et al., 2014), we can construct a modified version of QSGD (Alistarh et al., 2017) (special case of Shadowheart SGD from (Tyurin et al., 2024)), which we call Batch QSGD:

$$x^{k+1} = x^k - \frac{\gamma}{nbm}\sum_{i=1}^{n}\sum_{k=1}^{m}\mathcal{C}_{ik}\left(\sum_{j=1}^{b}\nabla f(x^k;\xi_{ij}^k)\right), \qquad \text{(Batch QSGD)}$$

where worker $i$ sends $m$ compressed vectors $\{\mathcal{C}_{ik}(\cdot)\}_{k\in[m]}$ to the server, which aggregates and calculates $x^{k+1}$. With $\text{Rand}K$ and proper parameters[1] (Tyurin et al., 2024), we can improve (2) to

$$\mathcal{O}\left(h\left(1 + \frac{\sigma^2}{n\varepsilon}\right)\frac{L\Delta}{\varepsilon} + \tau_{\text{w}}\left(\frac{d}{n} + 1\right)\frac{L\Delta}{\varepsilon} + \sqrt{\frac{d\tau_{\text{w}}h\sigma^2}{n\varepsilon}}\frac{L\Delta}{\varepsilon}\right). \qquad (3)$$

*Observation 2:* As in *Observation 1,* unlike (2), the time complexity (3) scales with the number of workers $n$, which once again justifies the use of many workers in the optimization of (1). The "statistical term" $h\sigma^2 L\Delta/n\varepsilon^2$ and the "communication term" $\tau_{\text{w}}dL\Delta/n\varepsilon$ improve linearly with $n$, while the "coupling term" $\sqrt{d\tau_{\text{w}}h\sigma^2/n\varepsilon}L\Delta/\varepsilon$ improves with the square root of $n$, which can reduce the effect of $d$ and $\sigma^2/\varepsilon$ for reasonably large $n$.

A high-level explanation for why the dependence on $d$ improves with $n$ is that all workers use i.i.d. and unbiased compressors $\{\mathcal{C}_{ik}\}$, which allow them to collaboratively explore more coordinates. This effect is similar to Synchronized SGD, where the variance $\mathbb{E}_\xi[\|\frac{1}{n}\sum_{i=1}^{n}\nabla f(x;\xi_i^k) - \nabla f(x)\|^2] \leq \frac{\sigma^2}{n}$ also improves with $n$. There are many other compressed methods that also improve with $n$, including DIANA (Mishchenko et al., 2019), Accelerated DIANA (Li et al., 2020), MARINA (Gorbunov et al., 2021), DASHA (Tyurin & Richtárik, 2023a), and FRECON (Zhao et al., 2021).

**3. Both communications can not be ignored.** Consider a more practical scenario, and our main point of interest, where the communication time from the server to the workers is $\tau_{\text{s}} > 0$. In this case, Batch QSGD requires

$$\mathcal{O}\left(h\left(1 + \frac{\sigma^2}{n\varepsilon}\right)\frac{L\Delta}{\varepsilon} + \tau_{\text{w}}\left(\frac{d}{n} + 1\right)\frac{L\Delta}{\varepsilon} + \sqrt{\frac{d\tau_{\text{w}}h\sigma^2}{n\varepsilon}}\frac{L\Delta}{\varepsilon} + \tau_{\text{s}}d\frac{L\Delta}{\varepsilon}\right) \qquad (4)$$

seconds because the server has to send $x^k \in \mathbb{R}^d$ of size $d$ to the workers in every iteration.

*Observation 3:* If $\tau_{\text{s}} \simeq \tau_{\text{w}}$, then (4) asymptotically equals $\mathcal{O}\left(h\left(L\Delta/\varepsilon + \sigma^2 L\Delta/n\varepsilon^2\right) + \tau_{\text{s}}dL\Delta/\varepsilon\right)$, reducing to (2), as in the method that does not compress at all! The "communication term" $\tau_{\text{s}}dL\Delta/\varepsilon$ **does not** improve with $n$.

We now arrive at our **main research question**:

> In the first case (**1. Communication is free**) and the second case (**2. Worker-to-server communication can not be ignored**), it is possible to design a method that scales the complexity with the number of workers $n$, while improving the dependencies on $d$ and $\sigma^2/\varepsilon$.
>
> Can we design a similarly efficient method for the third case (**3. Both communications can not be ignored**) using unbiased compressors, where the communication

---

[1] $b = \Theta(\frac{t^*}{h}), m = \Theta(\frac{t^*}{\tau_{\text{w}}}), t^* = \Theta\left(\max\left\{h, \tau_{\text{w}}, \frac{\tau_{\text{w}}d}{n}, \frac{h\sigma^2}{n\varepsilon}, \sqrt{\frac{d\tau_{\text{w}}h\sigma^2}{n\varepsilon}}\right\}\right), \gamma = \Theta(\frac{1}{L}), K = 1$ in $\text{Rand}K$

time from the server to the workers cannot be ignored, and where the dependence on both $d$ and $\sigma^2/\varepsilon$ improves with $n$, either linearly or with the square root of $n$?

At least, can this be achieved in the simplest homogeneous setting, where all workers have access to the same function, a scenario that arguably represents the simplest form of distributed optimization?

We know for certain that the answer is "No" in the *heterogeneous case*, due to the result of Gruntkowska et al. (2024), who proved that the iteration complexity does not improve with the number of workers $n$ under Assumptions 1.1 and 1.2. However, the homogeneous setting is "easier," giving us hope that the workers can exploit the fact that they all have access to the same distribution.

## 1.2 CONTRIBUTIONS

♠ **Lower bound.** Surprisingly, the answer is "No" to our **main research question**, even in the *homogeneous case*. We prove the following theorem.

**Theorem 1.6** (Informal Formulation of Theorems 4.2 and F.1). *Let Assumptions 1.1, 1.2, 1.3, and 1.4 hold. It is infeasible to find an $\varepsilon$–stationary point faster than*

$$\tilde{\Omega}\left(\min\left\{h\left(\frac{\sigma^2}{n\varepsilon}+1\right)\frac{L\Delta}{\varepsilon}+\tau_{\mathrm{w}}\left(\frac{d}{n}+1\right)\frac{L\Delta}{\varepsilon}+\sqrt{\frac{d\tau_{\mathrm{w}}h\sigma^2}{n\varepsilon}}\frac{L\Delta}{\varepsilon}+\tau_{\mathrm{s}}d\frac{L\Delta}{\varepsilon},h\frac{L\Delta}{\varepsilon}+h\frac{\sigma^2L\Delta}{\varepsilon^2}\right\}\right) \quad (5)$$

*seconds (up to logarithmic factors), using unbiased compressors (Def. 1.5) based on random sparsification, for all $L,\Delta,\varepsilon,n,\sigma^2,d,\tau_{\mathrm{w}},\tau_{\mathrm{s}},h>0$ such that $L\Delta\geq\tilde{\Theta}(\varepsilon)$ and dimension $d\geq\tilde{\Theta}\left(L\Delta/\varepsilon\right).$*

Because of the $\min$, the bound shows that it is possible to improve either the dependence on $d$ or the dependence on $\sigma^2/\varepsilon$ as the number of workers $n$ increases, but not both simultaneously. The lower bound is matched either by Batch QSGD or by the non-distributed SGD method (without any communication or cooperation). Moreover, if $\tau_{\mathrm{s}}\simeq\tau_{\mathrm{w}}$, the lower bound becomes $\tilde{\Omega}\left(\min\left\{h\left(\frac{\sigma^2}{n\varepsilon}+1\right)\frac{L\Delta}{\varepsilon}+\tau_{\mathrm{s}}d\frac{L\Delta}{\varepsilon},h\frac{L\Delta}{\varepsilon}+h\frac{\sigma^2L\Delta}{\varepsilon^2}\right\}\right)$, which can be matched by Batch Synchronized SGD with the complexity (2) (without compression) or by the non-distributed SGD method. In other words, if $\tau_{\mathrm{s}}\simeq\tau_{\mathrm{w}}$, then using methods with random sparsification compression in the distributed centralized setting offers no advantage. However, if $\tau_{\mathrm{s}}\lesssim\tau_{\mathrm{w}}$, the compression techniques can help on the workers side in the regimes when $\tau_{\mathrm{w}}d/n+\sqrt{d\tau_{\mathrm{w}}h\sigma^2/n\varepsilon}$ is larger than $\tau_{\mathrm{s}}d$, due to the former scaling with $n$.

♣ **New "worst-case" function.** To prove the lower bound, as we explain in Section 2.3, we needed a new "worst-case" function construction (see Section 3). We designed a new function $F_{T,K,a}$ in (9), which extends the ideas by Carmon et al. (2020). Proving its properties in Lemmas 3.1 and 3.2, as well as designing the function itself, can be an important contribution on its own.

♦ **Proof technique.** Using the new function, we develop a new proof technique and explain how the problem of establishing the lower bound reduces to a statistical problem (see Section 4), where we need to prove a concentration bound for a special sum (13), which represents the minimal possible random time required to find an $\varepsilon$–stationary point. Combining this result with the proven properties, we obtain our main result (11).

♥ **Improved analysis when $\tau_{\mathrm{w}}>0$.** To obtain the complete lower bound, we extended and improved the result by Tyurin et al. (2024), which was limited for our scenario and required additional modifications to finally obtain (5) (see Sections F and 5 for details).

## 2 PRELIMINARIES

For better comprehension of our new idea, we now present arguably one of the most important worst-case functions by Carmon et al. (2020), which is widely used to prove lower bounds in nonconvex optimization. It has been used by Arjevani et al. (2022; 2020a) to derive lower bounds in the stochastic setting, by Lu & De Sa (2021) in the decentralized setting, by Tyurin & Richtárik (2023b; 2024); Tyurin et al. (2024) in the asynchronous setting, by Huang et al. (2022) to show the lower iteration bound for unidirectional compressed methods, and by Li et al. (2021) in problems with a nonconvex-strongly-concave structure.

For any $T \in \mathbb{N}$, Carmon et al. (2020) define $F_T : \mathbb{R}^T \to \mathbb{R}$ such that [2]

$$F_T(x) := \sum_{i=1}^{T} \left[ \Psi(-x_{i-1})\Phi(-x_i) - \Psi(x_{i-1})\Phi(x_i) \right], \tag{6}$$

where $x_0 \equiv 1$, $x_i$ is the $i^{\text{th}}$ coordinate of $x \in \mathbb{R}^T$,

$$\Psi(x) = \begin{cases} 0, & x \leq 1/2, \\ \exp\left(1 - \frac{1}{(2x-1)^2}\right), & x \geq 1/2, \end{cases} \quad \text{and} \quad \Phi(x) = \sqrt{e} \int_{-\infty}^{x} e^{-\frac{1}{2}t^2} dt. \tag{7}$$

Notice that this function has a "chain–like" structure. If a method starts from $x^0 = 0$ and computes the gradient of $F_T$, then the gradient will have a non-zero value only in the first coordinate (use that $\Psi(0) = \Psi'(0) = 0$). Thus, by computing a single gradient, any "reasonable" method can "discover" at most one coordinate. At the same time, if the method wants to find an $\varepsilon$–stationary point, it should eventually discover the $T^{\text{th}}$ coordinate. These two facts imply that every "reasonable" method should compute the gradient of $F_T$ at least $T$ times. In the construction, Carmon et al. (2020) take $T = \Theta\left(\frac{L\Delta}{\varepsilon}\right)$. This construction is a "more technical" version of the celebrated quadratic optimization construction from (Nesterov, 2018), which has similar properties. Let us define $\text{prog}(x) := \max\{i \geq 0 \,|\, x_i \neq 0\}$ ($x_0 \equiv 1$), then the following lemma is a formalization of the described properties.

**Lemma 2.1** (Carmon et al. (2020)). *The function $F_T$ satisfies:*

1. *For all $x \in \mathbb{R}^T$, $\text{prog}(\nabla F_T(x)) \leq \text{prog}(x) + 1$.*

2. *For all $x \in \mathbb{R}^T$, if $\text{prog}(x) < T$, then $\|\nabla F_T(x)\| > 1$.*

Actually, in most proofs, the structure of (6) is not needed, and it is sufficient to work with Lemmas 2.1 and Lemma 2.2 from below, where the latter allows us to show that a scaled version of $F_T$ satisfies Assumptions 1.1 and 1.2.

**Lemma 2.2** (Carmon et al. (2020)). *The function $F_T$ satisfies:*

1. *$F_T(0) - \inf_{x \in \mathbb{R}^T} F_T(x) \leq \Delta^0 T$, where $\Delta^0 := 12$.*

2. *The function $F_T$ is $\ell_1$–smooth, where $\ell_1 := 152$.*

3. *For all $x \in \mathbb{R}^T$, $\|\nabla F_T(x)\|_\infty \leq \gamma_\infty$, where $\gamma_\infty := 23$.*

Hence, one of the main results by Carmon et al. (2020) was to show that it is infeasible to find an $\varepsilon$–stationary point without calculating $\mathcal{O}\left(\frac{L\Delta}{\varepsilon}\right)$ gradients of a function satisfying Assumptions 1.1 and 1.2. In turn, the classical gradient descent (GD) method matches this lower bound.

## 2.1 FAMILY OF DISTRIBUTED METHODS

In our lower bound, we focus on the family of methods described by Protocol 1. This protocol takes an algorithm as input and runs the standard functions of the workers and the server: the workers compute stochastic gradients locally, send compressed information, the server aggregates them asynchronously and in parallel, and sends compressed information back based on the local information. For now, we ignore the communication times from the workers to the server in Protocol 1.

For all $i \in [n]$, the algorithm can choose any point, based on the local information $I_i$, at which worker $i$ will start computing a stochastic gradient. It can also choose any point $s_i^k$, based on the server's local information $I$, along with the corresponding compressor $\mathcal{C}_i^k$, which will be sent to worker $i$. This protocol captures the behavior of virtually any asynchronous optimization process with workers connected to a server. We work with *zero-respecting* algorithms, as defined below.

**Definition 2.3.** We say that an algorithm $A$ that follows Protocol 1 is *zero-respecting* if it does not explore or assign non-zero values to any coordinate unless at least one of the available local vectors contains a non-zero value in that coordinate. The family of such algorithms we denote as $\mathcal{A}_{\text{zr}}$.

---

[2]similarly $F_T(x) := -\Psi(1)\Phi(x_1) + \sum_{i=2}^{T} \left[\Psi(-x_{i-1})\Phi(-x_i) - \Psi(x_{i-1})\Phi(x_i)\right]$ because $\Psi(-1) = 0$.

---

**Protocol 1**

---

1: **Input:** Algorithm $A \in \mathcal{A}_{zr}$
2: Init $I = \emptyset$ (all available information) on the server
3: Init $I_i = \emptyset$ (all available information) on worker $i$ for all $i \in [n]$
4: Run the following three loops in parallel. The first two on the workers. The third on the server.
5: **for** $i = 1, \ldots, n$ (in parallel on the workers) **do**
6:     **for** $k = 0, 1, \ldots$ **do**
7:         Algorithm $A$ calculates a new point $x$ based on local information $I_i$:    (takes 0 seconds)
        any vector $x \in \mathbb{R}^d$ such that $\mathrm{supp}(x) \in \cup_{y \in I_i}\mathrm{supp}(y)$    $(\mathrm{supp}(v) := \{\, i \in [d] : v_i \neq 0 \,\})$
8:         Calculate one stochastic gradient [3]$\nabla f(x;\xi), \quad \xi \sim \mathcal{D}_\xi$  ($\xi$ are i.i.d.)    (takes $h$ seconds)
9:         Add $\nabla f(x;\xi)$ to $I_i$    (takes 0 seconds)
10:     **end for**
11: **end for**
12: **for** $i = 1, \ldots, n$ (in parallel on the workers) **do**
13:     **for** $k = 0, 1, \ldots$ **do**
14:         Algorithm $A$ calculates a new point $\bar{s}_i^k$ based on local information $I_i$:    (takes 0 seconds)
        any vector $\bar{s}_i^k \in \mathbb{R}^d$ such that $\mathrm{supp}(\bar{s}_i^k) \in \cup_{y \in I_i}\mathrm{supp}(y)$
15:         Send $\bar{\mathcal{C}}_i^k(\bar{s}_i^k)$ to the server
        (takes $\tau_w \times \bar{P}_i^k$ seconds, where $\bar{P}_i^k$ is the number of coordinates retained by $\bar{\mathcal{C}}_i^k(\bar{s}_i^k)$)
16:         Add to $\bar{\mathcal{C}}_i^k(\bar{s}_i^k)$ to $I$    (takes 0 seconds)
17:     **end for**
18: **end for**
19: **for** $i = 1, \ldots, n$ (in parallel on the server) **do**
20:     **for** $k = 0, 1, \ldots$ **do**
21:         Algorithm $A$ calculates a new point $s_i^k$ based on local information $I$:    (takes 0 seconds)
        any vector $s_i^k \in \mathbb{R}^d$ such that $\mathrm{supp}(s_i^k) \in \cup_{y \in I}\mathrm{supp}(y)$
22:         Algorithm $A$ compresses the point: $\mathcal{C}_i^k(s_i^k) \quad \forall i \in [n]$    (takes 0 seconds)
23:         Send $\mathcal{C}_i^k(s_i^k)$ to the worker $i$
        (takes $\tau_s \times P_i^k$ seconds, where $P_i^k$ is the number of coordinates retained by $\mathcal{C}_i^k(s_i^k)$)
24:         Add to $\mathcal{C}_i^k(s_i^k)$ to $I_i$    (takes 0 seconds)
25:     **end for**
26: **end for**
    (a vector may be added to $I$ or $I_i$ at the same time as the algorithm calculates a new point; in this case, the protocol adds the vector first (with no delay since the operation takes 0 seconds))

---

This is the standard assumption (Carmon et al., 2020) that generalizes the family of methods working with the span of vectors (Nesterov, 2018) and holds for the majority of methods, including GD, Adam (Kingma & Ba, 2015), DORE (Liu et al., 2020), EF21-P (Gruntkowska et al., 2023), MARINA-P, and M3 (Gruntkowska et al., 2024).

### 2.2 Previous Lower Bound in the Heterogeneous Setting

Let us return back to our main question. In order to show that it is impossible to scale with $n$ in the *heterogeneous setting*, Gruntkowska et al. (2024) have proposed to use scaled versions of

$$G_j(x) := n \times \sum_{1 \leq i \leq T \text{ and } (i-1) \bmod n = j-1}^{T} [\Psi(-x_{i-1})\Phi(-x_i) - \Psi(x_{i-1})\Phi(x_i)]$$

for all $j \in [n]$, worker $i$ has access only a scaled version of $G_i$ for all $i \in [n]$. The idea is that the first block from (6) belongs to the first worker, the second block to the second worker, $\ldots$, and the $(n+1)^{\text{th}}$ block to the first worker again, and so on. Notice that $F_T(x) = \frac{1}{n}\sum_{i=1}^n G_i(x)$.

Notice one important property of this construction: only one worker at a time can discover the next coordinate. In other words, if the server sends a new iterate to all workers, only one worker, after computing the gradient, can make progress to the next coordinate.

The next step in (Gruntkowska et al., 2024), in the proof of the lower bound theorem, was to analyze Protocol 1. They consider[4] $\text{Rand}K$ with $K = 1$. Then, since the compressor sends only one coordinate with probability $p = 1/d$, the probability that the server sends the last non-zero coordinate to the worker responsible for the current block of (6) that can progress to the new coordinate is also $p$. Thus, the number of *consecutive* coordinates that the server has to send to the workers is at least $\sum_{j=1}^{T} \eta_j$, where $\eta_j$ is a geometric-like random variable with $\mathbb{P}(\eta_j = m | \eta_{j-1}, \ldots, \eta_1) \leq p(1 - p)^{m-1}$ for all $m \geq 1$. Using classical tools from statistical analysis, one can show that $\sum_{j=1}^{T} \eta_j \gtrsim T/p \simeq dL\Delta/\varepsilon$ with high probability. Thus, under Assumption 1.4, the communication time complexity cannot be better than $\Omega\left(\tau_{\text{s}} dL\Delta/\varepsilon\right)$, which does not improve with $n$.

## 2.3 FAILURE OF THE PREVIOUS CONSTRUCTION IN THE HOMOGENEOUS SETTING

However, in the *homogeneous* setting, if we want to reuse the idea, arguably the only option we have is to assign (scaled) $F_T$ to all workers to ensure that they all have the same function. But in this case, the arguments from Section 2.2 no longer apply, because all workers can simultaneously progress to the next coordinate, since they have access to all blocks of (6).

Indeed, if the server sends i.i.d. $\text{Rand}K$ compressors with $K = 1$, then the number of consecutive coordinates that the server has to send before the workers receive the last non-zero coordinate is $\sum_{j=1}^{T} \min_{i \in [n]} \eta_{ji}$, where $\mathbb{P}(\eta_{ji} = m | \{\eta_{kj}\}_{k<i}) \leq p(1-p)^{m-1}$ for all $m, j \geq 1, i \in [n]$. The $\min_{i \in [n]}$ operation appears because it is sufficient to wait for the first "luckiest" worker. Analyzing this sum, we can only show that

$$\tau_{\text{s}} \sum_{j=1}^{T} \min_{i \in [n]} \eta_{ji} \gtrsim \tau_{\text{s}} \frac{d}{n} \frac{L\Delta}{\varepsilon}, \tag{8}$$

with high probability, which scales with $n$ due to $\min$.

There are two options: either $\Omega(\tau_{\text{s}} dL\Delta/n\varepsilon)$ is tight and it is possible to find a method that matches it, or we need to find another way to improve the lower bound. To prove the latter, we arguably need a different fundamental construction from (6), which we propose in the next section.

## 3 A NEW "WORST-CASE" FUNCTION

In this section, we give a less technical description of our lower bound construction and the main theorem from Section D. Instead of (6), we propose to use another "worst-case" function. For any $T, K \in \mathbb{N}$, and $e \geq a > 1$, we define the function $F_{T,K,a} : \mathbb{R}^T \to \mathbb{R}$ such that

$$F_{T,K,a}(x) = -\sum_{i=1}^{T} \Psi_a(x_{i-K}) \ldots \Psi_a(x_{i-2}) \Psi_a(x_{i-1}) \Phi(x_i) + \sum_{i=1}^{T} \Gamma(x_i), \tag{9}$$

$$\Psi_a(x) = \begin{cases} 0, & x \leq 1/2, \\ \exp\left(\log a \cdot \left(1 - \frac{1}{(2x-1)^2}\right)\right), & x > 1/2, \end{cases} \qquad \Phi(x) = \sqrt{e} \int_{-\infty}^{x} e^{-\frac{1}{2}t^2} dt, \tag{10}$$

$$\Gamma(x) = \begin{cases} -xe^{1/x+1}, & x < 0, \\ 0, & x \geq 0, \end{cases}$$

---

[3]i) Multiple queries with the same random variable do not change the lower bound; see Remark E.1 in Section E; ii) In the heterogeneous setup (Section 2.2), worker $i$ computes $\nabla f_i(x; \xi)$, where $f_i$ is its local function.

[4]In general, they presented a more general setting where the server can zero out coordinates with any probability, capturing not only $\text{Rand}K$ with $K = 1$ and $p = 1/d$, but also $\text{Rand}K$ with $K > 1$ and other compressors.

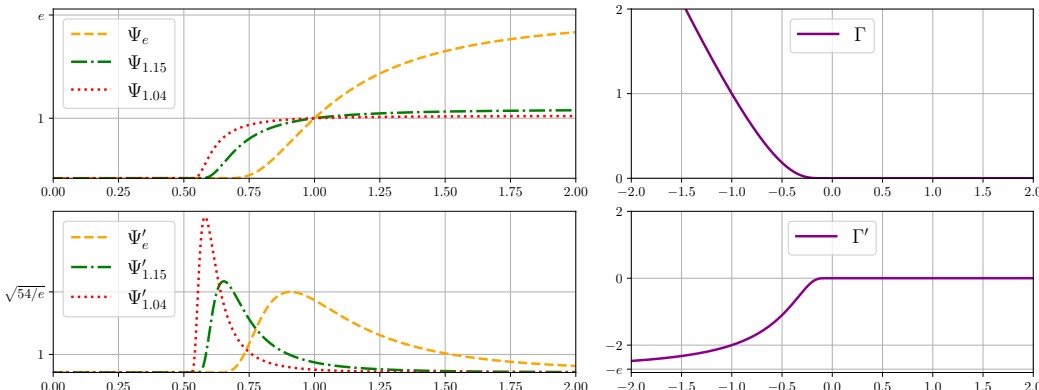

Figure 1: The functions $\Psi_a(x)$ and $\Gamma(x)$, along with their derivatives $\Psi_a'(x)$ and $\Gamma'(x)$. The plots of $\Phi(x)$ and $\Phi'(x)$ are shown in (Carmon et al., 2020).

and $x_0 = \cdots = x_{-K+1} \equiv 1$. The main modification is that instead of the block $-\Psi(x_{i-1})\Phi(x_i)$, we use $-\Psi_a(x_{i-K})\ldots\Psi_a(x_{i-2})\Psi_a(x_{i-1})\Phi(x_i)$ (ignore $a$ for now). In the previous approach, it was sufficient for a worker to have $x_{i-1} \neq 0$ to discover the next $i^{\text{th}}$ coordinate. However, in our new construction, the worker needs $x_{i-1} \neq 0, x_{i-2} \neq 0, \ldots, x_{i-K+1} \neq 0$ for that. With this modification, it is not sufficient for the "luckiest" worker to get the non-zero $i-1^{\text{th}}$ coordinate to discover the next coordinate: to progress to the $i_{\text{th}}$ coordinate, the worker should also have non-zero $i-2^{\text{th}}, \ldots, i-K+1^{\text{th}}$ coordinates.

Next, we remove the positive blocks $\Psi(x_{i-1})\Phi(x_i)$, which we believe was introduced to prevent the methods from ascending, exploring negative values of $x_i$, and finding a nearby stationary point "above." Instead, we introduce $\Gamma(x_i)$, which serves the same purpose: if a method starts exploring negative values, this term prevents it from reaching a stationary point there. Let us define

$$\text{prog}^K(x) := \max\{i \geq 0 \,|\, x_i \neq 0, x_{i-1} \neq 0, \ldots, x_{i-K+1} \neq 0\}.$$

Instead of Lemma 2.1, we prove the following lemma:

**Lemma 3.1** (Lemmas D.4 and D.5). *The function $F_{T,K,a}$ satisfies:*

1. *For all $x \in \mathbb{R}^T$, $\text{supp}(\nabla F_{T,K,a}(x)) \subseteq \{1, \ldots, \text{prog}^K(x) + 1\} \cup \text{supp}(x)$, where $\text{supp}(v) := \{\, i \in [d] : v_i \neq 0 \,\}$.*

2. *For all $x \in \mathbb{R}^T$, if $\text{prog}^K(x) < T$, then $\|\nabla F_{T,K,a}(x)\| > 1$.*

The function $F_{T,K,a}$ remains smooth. However, by multiplying with additional $\Psi$ terms, we alter its geometry and make it more chaotic: the difference $F_{T,K,a}(0) - \inf_{x \in \mathbb{R}^T} F_{T,K,a}(x)$, the smoothness constant, and the maximum $\ell_\infty$–norm may increase. To mitigate this, we introduce the parameter $a$ in (10) that allows us to control these properties. Notice that if $a = e$, then $\Psi_a(x) = \Psi(x)$ for all $x \in \mathbb{R}^T$, where $\Psi$ is defined in (7). Instead of Lemma 2.2, we prove

**Lemma 3.2** (Lemmas D.6, D.7, and D.8). *The function $F_{T,K,a}$ satisfies:*

1. *$F_{T,K,a}(0) - \inf_{x \in \mathbb{R}^T} F_{T,K,a}(x) \leq \Delta^0(K, a) \cdot T$, where $\Delta^0(K, a) := \sqrt{2\pi e} \cdot a^K$.*

2. *The function $F_{T,K,a}$ is $\ell_1(K, a)$–smooth, where $\ell_1(K, a) := 12\sqrt{2\pi}e^{5/2} \cdot \frac{K^2 a^K}{\log a}$.*

3. *For all $x \in \mathbb{R}^T$, $\|\nabla F_{T,K,a}(x)\|_\infty \leq \gamma_\infty(K, a)$, where $\gamma_\infty(K, a) := 6\sqrt{2\pi}e^{3/2} \cdot \frac{Ka^K}{\sqrt{\log a}}$.*

Taking $K = 1$ and $a = e$, up to constant factors, Lemmas 3.1 and 3.2 reduce to Lemmas 2.1 and 2.2. The larger the value of $K$, the larger the bounds in Lemma 3.2, and this growth can be exponential if $a = e$. However, with a proper choice of $1 < a \ll e$, we can mitigate the increase caused by $K$.

## 4 LOWER BOUND WITH SERVER-TO-WORKER (S2W) COMMUNICATION

We now present informal and formal versions of our main result:

**Theorem 4.1** (Informal Formulation of Theorem 4.2). *Let Assumptions 1.1, 1.2, 1.3, and 1.4 hold. It is infeasible to find an $\varepsilon$–stationary point faster than*

$$\tilde{\Omega}\left(\min\left\{\tau_s d \frac{L\Delta}{\varepsilon}, h\frac{L\Delta}{\varepsilon} + h\frac{\sigma^2 L\Delta}{\varepsilon^2}\right\}\right) \tag{11}$$

*seconds (up to logarithmic factors), using unbiased compressors (Def. 1.5) based on random sparsification, for all $L, \Delta, \varepsilon, n, \sigma^2, d, \tau_s, h > 0$ such that $L\Delta \geq \tilde{\Theta}(\varepsilon)$ and dimension $d \geq \tilde{\Theta}(L\Delta/\varepsilon)$.*

**Theorem 4.2.** *Let $L, \Delta, \varepsilon, n, \sigma^2, d, \tau_s, \tau_w, h > 0$ be any numbers such that $\bar{c}_1 \varepsilon \log^4(n+1) < L\Delta$ and dimension $d \geq \bar{c}_3 \frac{L\Delta}{\log^3(n+1)\varepsilon}$. Consider Protocol 1. For all $i \in [n]$ and $k \geq 0$, compressor $\mathcal{C}_i^k$ selects and transmits $P_i^k$ uniformly random coordinates without replacement, scaled by any constants[5], where $P_i^k \in \{0, \ldots, d\}$ may vary across each compressor [6]. Then, for any algorithm $A \in \mathcal{A}_{zr}$ (Def. 2.3), there exists a function $f : \mathbb{R}^d \to \mathbb{R}$ such that $f$ is L-smooth, i.e., $\|\nabla f(x) - \nabla f(y)\| \leq L \|x - y\|$ for all $x, y \in \mathbb{R}^d$, and $f(0) - \inf_{x \in \mathbb{R}^d} f(x) \leq \Delta$, exists a stochastic gradient oracles that satisfies Assumption 1.3, and $\mathbb{E}\left[\inf_{y \in S_t} \|\nabla f(y)\|^2\right] > \varepsilon$ for all*

$$t \leq \bar{c}_2 \times \left(\frac{1}{\log^3(n+1)} \cdot \frac{L\Delta}{\varepsilon}\right)\min\left\{\frac{1}{\log(n+1)} \cdot \tau_s d, \max\left\{h, \frac{1}{\log^3(n+1)} \cdot \frac{h\sigma^2}{\varepsilon}\right\}\right\}, \tag{12}$$

*where $S_t$ is the set of all possible points that can be constructed by A up to time t based on I and $\{I_i\}$. The quantities $\bar{c}_1, \bar{c}_2,$ and $\bar{c}_3$ are universal constants.*

The formulation of Theorem 4.2 is standard in the literature. However, following Tyurin & Richtárik (2023b), we present the lower bound in terms of *time complexities* rather than *iteration complexities*. Then, following Huang et al. (2022); He et al. (2023); Tyurin et al. (2024), we consider a subfamily of unbiased compressors based from Definition 1.5 on random sparsification to prove the lower bound; this is standard practice for taking the "worst-case" compressors from the family (similarly to taking the "worst-case" functions (Carmon et al., 2020; Nesterov, 2018)). Moreover, due to the uncertainty principle (Safaryan et al., 2022), all unbiased compressors exhibit variance and communication cost comparable to those of the RandK sparsifier in the worst case (up to constant factors).

The main observation in Theorems 4.1 and 4.2 is that it is not possible to scale both $d$ and $\sigma^2/\varepsilon$ by more than $\log^4(n + 1)$ and $\log^6(n + 1)$, respectively. Asymptotically, this scaling is significantly worse than the linear $n$ and square-root $\sqrt{n}$ scalings discussed in Section 1.1. For instance, if $n = 10,000$ and $d$ is increased by a factor of 10, we have to increase $n$ by a factor of $10^3$ (two factors more) to ensure that $\tau_s d/\log^4(n+1)$ does not change.

In Section A, we present the intuition and the proof sketch of the result.

## 5 LOWER BOUND WITH BOTH W2S AND S2W COMMUNICATION

In the previous section, we provide the lower bound without taking into account the communication cost $\tau_w$. Combining Theorem 4.2 with our new Theorem F.1, which extends the result by Tyurin et al. (2024) for our setup, we can obtain the complete lower bound (5) from Theorem 1.6 with $\tau_w > 0$ and $\tau_s > 0$. Notice that if $\tau_s \simeq \tau_w$, then the lower bound is $\tilde{\Omega}\left(\min\left\{h\left(\frac{\sigma^2}{n\varepsilon} + 1\right)\frac{L\Delta}{\varepsilon} + \tau_s d\frac{L\Delta}{\varepsilon}, h\frac{L\Delta}{\varepsilon} + h\frac{\sigma^2 L\Delta}{\varepsilon^2}\right\}\right)$. Up to logarithmic factors, under Assumptions 1.1, 1.2, 1.3, and 1.4, it is infeasible to improve both $d$ and $\sigma^2/\varepsilon$ as $n$ increases.

### 5.1 ALGORITHMS ALMOST MATCHING THE LOWER BOUND

Due to the $\min$, there are two regimes in which the lower bound (5) operates. If the second term is smaller in (5), then the lower bound is $\tilde{\Omega}\left(\frac{hL\Delta}{\varepsilon} + \frac{h\sigma^2 L\Delta}{\varepsilon^2}\right)$, which is matched by the vanilla

---

[5]To potentially preserve unbiasedness. For instance, RandK scales by $d/K$.

[6]For instance, the compressors can be RandK (see Def. C.1) with any $K \in [d]$, PermK (Szlendak et al., 2021), Identity compressor when $P_i^k = d$.

SGD method run locally (without any communication or cooperation). Otherwise, if the first term is smaller, then the lower bound is matched by Batch QSGD, which has the matching complexity (4) (up to logarithmic factors). Moreover, in the latter case, if $\tau_{\mathrm{s}} \simeq \tau_{\mathrm{w}}$, the lower bound becomes $\tilde{\Omega}\left(\min\left\{h\left(\frac{\sigma^2}{n\varepsilon}+1\right)\frac{L\Delta}{\varepsilon}+\tau_{\mathrm{s}}d\frac{L\Delta}{\varepsilon}\right\}\right)$, which can be matched by Batch Synchronized SGD with the complexity (2); thus, if $\tau_{\mathrm{s}} \simeq \tau_{\mathrm{w}}$, then unbiased sparsified compression is not needed at all, as it cannot help due to the lower bound.

## 6 Conclusion

We prove nearly tight lower bounds for centralized distributed optimization under the computation and communication Assumption 1.4. We show that *even in the homogeneous scenario*, it is not possible to scale both $d$ and $\sigma^2/\varepsilon$ by more than poly-logarithmic factors in $n$. Notice that the family of unbiased compressors contains the family of biased compressors (Beznosikov et al., 2020). Therefore, our lower bounds also apply to methods that use biased compressors, in the sense that there exists a "worst-case" compressor for which these methods cannot achieve a convergence rate faster than the lower bound in Theorem 1.6.

The lower bounds are tight only up to logarithmic factors. Thus, a possible challenging direction is to improve the powers of the logarithms, or even eliminate the logarithms entirely. The latter (if at all possible) may be very challenging and would likely require entirely different constructions and techniques. Another limitation is that the lower bounds are constructed using random sparsifiers. Due to the uncertainty principle (Safaryan et al., 2022), we conjecture that the bounds also hold for the entire family of unbiased compressors, but proving this would require more sophisticated constructions.

In practice, however, biased and unbiased compressors, including Top$K$ and Rank$K$ (Alistarh et al., 2018; Vogels et al., 2019), exhibit significantly better compression properties than those predicted by worst-case analysis (Beznosikov et al., 2020). When used on the server side in combination with EF or EF21-P (Gruntkowska et al., 2023; Tyurin et al., 2024), they may help mitigate the pessimistic term $\tau_{\mathrm{s}}d^{L\Delta}/\varepsilon$. Moreover, our pessimistic lower bound may potentially be broken under additional assumptions such as convexity or second-order smoothness.

## Acknowledgements

The work was supported by the grant for research centers in the field of AI provided by the Ministry of Economic Development of the Russian Federation in accordance with the agreement 000000C313925P4F0002 and the agreement №139-10-2025-033.

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

CONTENTS

## A    PROOF SKETCH

We illustrate the main idea behind the proof and how the new "worst-case" function helps to almost eliminate the scaling with $n$. Consider the first $K$ coordinates of $F_{T,K,a}$ (which is scaled in the proof to satisfy Assumptions 1.1 and 1.2). Recall that, due to Lemma 3.1, the only way to discover the $K+1^{\text{th}}$ coordinate in any worker is to ensure that all of the first $K$ coordinates are non-zero.

**Reduction to a statistical problem.** There are only two options by which a worker may discover a new non-zero coordinate: through local stochastic computations or through communication from the server. In the first option, a worker computes a stochastic gradient, which takes $h$ seconds. However, due to the construction of stochastic gradients (Arjevani et al., 2022), even if the computation is completed, the worker will not make progress or discover a new non-zero coordinate, as it will be zeroed out with probability $p_\sigma = \Theta\left(\varepsilon \cdot \gamma_\infty^2(K,a)/\sigma^2\right)$. In the second option, due to the condition of Theorem 4.2, a worker receives a stream of uniformly sampled coordinates $\nu_1, \nu_2, \ldots$ (workers get different streams), and the worker can discover a new non-zero coordinate only if random variable $\nu_i \in [K]$, which satisfies $\mathbb{P}\left(\nu_i \in [K]|\nu_1,\ldots,\nu_{i-1}\right) \leq {}^K/_{T-i+1} \leq p_K := {}^{2K}/_T$ for all $i \leq {}^T/_2$.

Next, we define two sets of random variables: (i) let $\eta_{1,i,k}$ denote the number of stochastic gradient computations until the first moment when a coordinate is not zeroed out in the stochastic gradient oracle (see (26)), after the moment when the $(k-1)^{\text{th}}$ coordinate is no longer zeroed out in worker $i$; (ii) let $\mu_{1,i,k}$ be the number of received coordinates until the moment when the last received coordinate belongs to $[K]$, after the $(k-1)^{\text{th}}$ time this has happened. In other words, $\eta_{1,i,1}$ is the number of stochastic gradient computations until the moment when the algorithm receives a "lucky" stochastic gradient where the last coordinate is not zeroed out. The random variable $\eta_{1,i,2}$ is the number of computations until it happens for the second time, and so on. Similarly, $\mu_{1,i,1}$ is the position of the first coordinate from the stream sent by the server to worker $i$ that belongs to $[K]$. The random variable $\mu_{1,i,2}$ refers to the second time this occurs, and so on. By definition, the sequences $\{\eta_{1,i,k}\}$ and $\{\mu_{1,i,k}\}$ follow *approximately* geometric-like distributions with parameters $p_\sigma$ and $p_K$, respectively.

To discover all of the first $K$ coordinates, either the first or the second process must uncover at least ${}^K/_2$ coordinates. If worker $i$ has discovered fewer than ${}^K/_2$ coordinates through stochastic gradient computations, and fewer than ${}^K/_2$ coordinates through receiving them from the server, then it will not be able to cover all $K$ coordinates. Thus, the algorithm should wait at least $\min_{i \in [n]} \left\{ \min \left\{ h \sum_{k=1}^{\frac{K}{2}} \eta_{1,i,k}, \tau_s \sum_{k=1}^{\frac{K}{2}} \mu_{1,i,k} \right\} \right\}$ seconds until the moment when it can potentially discover the $K+1^{\text{th}}$ coordinate, where the outer minimum $\min_{i \in [n]}$ appears because it is sufficient for the algorithm to wait for the first "luckiest" worker. Repeating the same arguments $B := \lfloor {}^T/_K \rfloor$ times, the algorithms requires at least

$$t_B := \sum_{b=1}^{B} \min_{i \in [n]} \left\{ \min \left\{ h \sum_{k=1}^{K/2} \eta_{b,i,k}, \tau_s \sum_{k=1}^{K/2} \mu_{b,i,k} \right\} \right\} \tag{13}$$

seconds to discover the $T^{\text{th}}$ coordinate and potentially find an $\varepsilon$–stationary point, where the sequences $\{\eta_{b,i,k}\}$ and $\{\mu_{b,i,k}\}$ follow *approximately* geometric-like distributions with $p_\sigma$ and $p_K$, respectively.

**Analysis of the concentration.** Hence, we have reduced the lower bound to the analysis of the sum $t_B$. Recall (8), where the lower bound improves with $n$ due to $\min_{i \in [n]}$. In (13), we also get $\min_{i \in [n]}$. However, and this is the main reason for the new construction, there are two sums $\sum_{k=1}^{K/2}$, which allows us to mitigate the influence of the $\min_{i \in [n]}$. In particular, we can show that $t_B \gtrsim \frac{BK}{n^{1/K}} \min\{h/p_\sigma, \tau_s/p_K\}$ with high probability. Notice that the first fraction improves with $n^{\frac{1}{K}}$ instead of $n$ due to the sums; thus, the larger $K$, the smaller the influence of $n$.

**Putting it all together.** However, we cannot take $K$ too large due to Lemma 3.2. Substituting the choice of $T$, $p_\sigma$, and $p_K$ (defined in the proof of Theorem 4.2 to ensure that Assumptions 1.1, 1.2, and 1.3 are satisfied and the scaled version of $F_{T,K,a}$ has the squared norm larger than $\varepsilon$ while the $T^{\text{th}}$ is not discovered), we can show that

$$t_B \gtrsim \frac{L\Delta}{n^{1/K} \cdot \Delta^0(K,a) \cdot \ell_1(K,a) \cdot \varepsilon} \min \left\{ \max \left\{ h, \frac{h\sigma^2}{\varepsilon \cdot \gamma_\infty^2(K,a)} \right\}, \frac{\tau_s d}{K} \right\},$$

with high probability, where $\Delta^0(K,a)$, $\ell_1(K,a)$, and $\gamma_\infty(K,a)$ are defined in Lemma 3.2. The final step is to choose $K = \Theta\left(\log n\right)$ and $a = 1 + {}^1/_K$ to obtain the result of Theorem 4.2.

## B  ADDITIONAL RELATED WORK

While we focus on lower bounds in the context of stochastic optimization and compressed vectors in nonconvex settings, there is much related work in other domains and setups. The seminal works on lower bounds were done by Nemirovskij & Yudin (1983); Nesterov (2018), where Nesterov (2018) showed that the accelerated gradient descent (Nesterov, 1983) is optimal in the convex setting using a quadratic "worst-case" function. In the nonconvex setting, Carmon et al. (2020) provided an alternative function, described in the main part of the paper. For convex problems, Woodworth et al. (2018) introduced the graph oracle, a generalization of the classical gradient oracle (Nemirovskij & Yudin, 1983; Nesterov, 2018), and established lower bounds for a broad class of parallel optimization methods. Arjevani et al. (2020b) further analyzed the delayed gradient descent method, which corresponds to Asynchronous SGD with constant iteration delays. Tyurin & Richtárik (2023b; 2024); Tyurin et al. (2024) proved lower bounds for methods in asynchronous settings. Fang et al. (2018); Patel et al. (2022) studied a different setting from Assumption 1.3, where they assumed the mean-squared smoothness property to enable the analysis of methods with variance reduction techniques (Fang et al., 2018; Cutkosky & Orabona, 2019). Woodworth & Srebro (2016) considered the finite-sum setting in the convex setting. Woodworth et al. (2020; 2021) proved that the min-max optimal algorithm for optimizing smooth convex objectives in the intermittent communication setting is the best of accelerated local and minibatch SGD, which leads to a similar conclusion to ours; however, their results are related to, but not directly comparable with ours, since we analyze the limited scalability of improving both stochastic noise and communication complexity through compressors. Glasgow et al. (2022) provided sharp lower bounds for local SGD approaches in terms of iteration complexity. Huang et al. (2022); He et al. (2023); Gruntkowska et al. (2024) provided lower bounds for compression techniques, but in the heterogeneous setting.

## C  AUXILIARY FACTS AND NOTATIONS

**Definition C.1** (RandK). Assume that $S$ is a random subset of $[d]$ such that $|S| = K$ for some $K \in [d]$. A stochastic mapping $\mathcal{C} : \mathbb{R}^d \times \mathbb{S}_\nu \to \mathbb{R}^d$ is called RandK if

$$\mathcal{C}(x; S) = \frac{d}{K} \sum_{j \in S} x_j e_j,$$

where $\{e_i\}_{i=1}^d$ denotes the standard unit basis. The set $S$ can be produced with a uniform sampling of $[d]$ without replacement.

### C.1  NOTATIONS

$\mathbb{N} := \{1, 2, \dots\}$; $\|x\|$ is the output of the standard Euclidean norm for all $x \in \mathbb{R}^d$; $\langle x, y \rangle = \sum_{i=1}^d x_i y_i$ is the standard dot product; $\|A\|$ is the standard spectral/operator norm for all $A \in \mathbb{R}^{d \times d}$; $g = \mathcal{O}(f)$ : exist $C > 0$ such that $g(z) \leq C \times f(z)$ for all $z \in \mathcal{Z}$; $g = \Omega(f)$ : exist $C > 0$ such that $g(z) \geq C \times f(z)$ for all $z \in \mathcal{Z}$; $g = \Theta(f)$ : $g = \mathcal{O}(f)$ and $g = \Omega(f)$; $g = \widetilde{\mathcal{O}}(f), g = \widetilde{\Omega}(f), g = \widetilde{\Theta}(f)$ : the same as $g = \mathcal{O}(f), g = \Omega(f), g = \Theta(f)$, respectively, but up to logarithmic factors; $g \simeq h$ : $g$ and $h$ are equal up to universal positive constants; $g \gtrsim h$ : $g$ greater or equal to $h$ up to universal positive constants; $\mathcal{C}$ is an unbiased compressor (Definition 1.5); $\text{supp}(v) = \{i \in [d] : v_i \neq 0\}$; $h$ : maximum time (in seconds) for any worker to compute one stochastic gradient; $\tau_\text{s}$ : communication time per coordinate from the server to any worker; $\tau_\text{w}$ : communication time per coordinate from any worker to the server;

## D  LOWER BOUND

### D.1  NEW CONSTRUCTION

For any $T, K \in \mathbb{N}$, and $e \geq a > 1$ we define the function $F_{T,K,a} : \mathbb{R}^T \to \mathbb{R}$ such that

$$F_{T,K,a}(x) = -\sum_{i=1}^T \Psi_a(x_{i-K}) \dots \Psi_a(x_{i-2}) \Psi_a(x_{i-1}) \Phi(x_i) + \sum_{i=1}^T \Gamma(x_i), \tag{14}$$

where $x_i$ is the $i^{\text{th}}$ coordinate of a vector $x \in \mathbb{R}^T$ and

$$\Psi_a(x) = \begin{cases} 0, & x \leq 1/2, \\ \exp\left(\log a \cdot \left(1 - \frac{1}{(2x-1)^2}\right)\right), & x > 1/2, \end{cases} \qquad \Phi(x) = \sqrt{e} \int_{-\infty}^{x} e^{-\frac{1}{2}t^2} dt,$$

and

$$\Gamma(x) = \begin{cases} -xe^{1/x+1}, & x < 0, \\ 0, & x \geq 0. \end{cases}$$

We assume that $x_0 = \cdots = x_{-K+1} \equiv 1$. Importantly, throughout the lower bound analysis, we assume that $e \geq a > 1$, even if this assumption is not explicitly stated in all theorems.

We additionally define

$$\text{prog}^K(x) := \max\{i \geq 0 \,|\, x_i \neq 0, x_{i-1} \neq 0, \ldots, x_{i-K+1} \neq 0\}$$
$$(x_0 = \cdots = x_{-K+1} \equiv 1),$$

which extends $\text{prog}(x) \equiv \text{prog}^1(x) := \max\{i \geq 0 \,|\, x_i \neq 0\}$ $(x_0 \equiv 1)$.

## D.2 Auxiliary Lemmas

In this section, we list useful properties of the functions $\Phi$, $\Gamma$, $\Psi_a$, and $F_{T,K,a}$. We prove them in Section D.3.

**Lemma D.1** (Carmon et al. (2020)). *Function $\Phi$ is twice differentiable and satisfies*

$$0 \leq \Phi(x) \leq \sqrt{2\pi e}, \quad 0 \leq \Phi'(x) \leq \sqrt{e}, \text{ and } |\Phi''(x)| \leq 27$$

*for all $x \in \mathbb{R}$. Moreover, $\Phi'(x) > 1$ for all $-1 < x < 1$.*

**Lemma D.2.** *Function $\Gamma$ is twice differentiable and satisfies*

$$0 \leq \Gamma(x), \quad -e < \Gamma'(x) \leq 0, \text{ and } 0 \leq \Gamma''(x) \leq 27e^{-2}$$

*for all $x \in \mathbb{R}$. Moreover, $\Gamma'(x) \leq -2$ for all $x \leq -1$.*

**Lemma D.3.** *Function $\Psi_a$ is twice differentiable and satisfies*

$$0 \leq \Psi_a(x) < a, \quad 0 \leq \Psi_a'(x) \leq \frac{2e}{\sqrt{\log a}}, \text{ and } |\Psi_a''(x)| \leq \frac{56e}{\log a}$$

*for all $x \in \mathbb{R}$ and $1 < a \leq e$. Moreover, $\Psi_a(x) \geq 1$ for all $x \geq 1$ and $1 < a \leq e$.*

**Lemma D.4.** *For all $x \in \mathbb{R}^T$, $\text{supp}(\nabla F_{T,K,a}(x)) \subseteq \{1, \ldots, \text{prog}^K(x) + 1\} \cup \text{supp}(x)$, where $\text{supp}(v) := \{ i \in [d] : v_i \neq 0 \}$.*

**Lemma D.5.** *For all $x \in \mathbb{R}^T$, if $\text{prog}^K(x) < T$, then $\|\nabla F_{T,K,a}(x)\| > 1$.*

**Lemma D.6.** *Function $F_{T,K,a}$ satisfies*

$$F_{T,K,a}(0) - \inf_{x \in \mathbb{R}^T} F_{T,K,a}(x) \leq \Delta^0(K,a) \cdot T,$$

*where $\Delta^0(K,a) := \sqrt{2\pi e} \cdot a^K$.*

**Lemma D.7.** *For all $x \in \mathbb{R}^T$, $\|\nabla F_{T,K,a}(x)\|_\infty \leq \gamma_\infty(K,a)$, where $\gamma_\infty(K,a) := 6\sqrt{2\pi}e^{3/2} \cdot \frac{Ka^K}{\sqrt{\log a}}$.*

**Lemma D.8.** *The function $F_{T,K,a}$ is $\ell_1(K,a)$–smooth, i.e., $\|\nabla^2 F_{T,K,a}(x)\| \leq \ell_1(K,a)$ for all $x \in \mathbb{R}^T$, where $\ell_1(K,a) := 12\sqrt{2\pi}e^{5/2} \cdot \frac{K^2 a^K}{\log a}$.*

### D.3 Proof of Lemmas

**Lemma D.2.** *Function $\Gamma$ is twice differentiable and satisfies*

$$0 \le \Gamma(x), \quad -e < \Gamma'(x) \le 0, \text{ and } 0 \le \Gamma''(x) \le 27e^{-2}$$

*for all $x \in \mathbb{R}$. Moreover, $\Gamma'(x) \le -2$ for all $x \le -1$.*

*Proof.* The first fact is due to $\lim\limits_{\Delta \to 0} \frac{\Gamma(\Delta)}{\Delta} = 0$, $\Gamma'(0) = 0$, and $\lim\limits_{\Delta \to 0} \frac{\Gamma'(\Delta)}{\Delta} = 0$. $\Gamma$ is clearly non-negative. Next, for all $x \le 0$,

$$\Gamma'(x) = -e^{1/x+1} + \frac{e^{1/x+1}}{x}$$

and

$$\Gamma''(x) = -\frac{e^{1/x+1}}{x^3}.$$

Thus, $\Gamma'$ is strongly increasing for all $x \le 0$, and $\lim\limits_{x \to -\infty} \Gamma'(x) = -e < \Gamma'(x) \le 0$. Next, $\Gamma''(x) \ge 0$ for all $x \le 0$, and $\max\limits_{x \le 0} \Gamma''(x) = 27e^{-2}$ for all $x \le 0$. $\qquad\square$

**Lemma D.3.** *Function $\Psi_a$ is twice differentiable and satisfies*

$$0 \le \Psi_a(x) < a, \quad 0 \le \Psi_a'(x) \le \frac{2e}{\sqrt{\log a}}, \text{ and } |\Psi_a''(x)| \le \frac{56e}{\log a}$$

*for all $x \in \mathbb{R}$ and $1 < a \le e$. Moreover, $\Psi_a(x) \ge 1$ for all $x \ge 1$ and $1 < a \le e$.*

*Proof.* The differentiability at $x = \frac{1}{2}$ follows from $\lim\limits_{\Delta \to 0} \frac{\Psi_a(\frac{1}{2}+\Delta)}{\Delta} = 0$ for all $a > 1$. For all $x \le \frac{1}{2}$, $\Psi_a'(x) = 0$. For all $x > \frac{1}{2}$, we get

$$0 \le \Psi_a'(x) = \frac{4 \log a}{(2x-1)^3} \exp\left(\log a \left(1 - \frac{1}{(2x-1)^2}\right)\right)$$

$$= \frac{4a}{\sqrt{\log a}} \times \frac{\log^{3/2} a}{(2x-1)^3} \exp\left(-\frac{\log a}{(2x-1)^2}\right).$$

Taking $t = \frac{\log^{1/2} a}{(2x-1)} > 0$ and using $t^3 e^{-t^2} \le \frac{1}{2}$, we get

$$\Psi_a'(x) \le \frac{4a}{\sqrt{\log a}} \times \frac{1}{2} \le \frac{2e}{\sqrt{\log a}}$$

since $a \le e$.

Clearly, $\Psi_a(x) \ge 0$ for all $x \in \mathbb{R}$, and $\Psi_a$ is non-decreasing. Moreover it is strongly monotonic for all $x > \frac{1}{2}$. Thus $\Psi_a(x) < \lim\limits_{x \to \infty} \Psi_a(x) = a$ for all $x \in \mathbb{R}$.

The twice differentiability at $x = \frac{1}{2}$ follows from $\lim\limits_{\Delta \to 0} \frac{\Psi_a'(\frac{1}{2}+\Delta)}{\Delta} = 0$ for all $a > 1$. For all $x \le \frac{1}{2}$, $\Psi_a''(x) = 0$. For all $x > \frac{1}{2}$, taking the second derivative and using simple algebra, we get

$$|\Psi_a''(x)| = \left| -\frac{8 \log a \times (3(2x-1)^2 - 2 \log a)}{(2x-1)^6} \exp\left(\log a \left(1 - \frac{1}{(2x-1)^2}\right)\right) \right|$$

$$= \left| \frac{8a \log a \times (3(2x-1)^2 - 2 \log a)}{(2x-1)^6} \exp\left(-\frac{\log a}{(2x-1)^2}\right) \right|$$

$$\le \left| \frac{24a \log a}{(2x-1)^4} \exp\left(-\frac{\log a}{(2x-1)^2}\right) \right| + \left| \frac{16a \log^2 a}{(2x-1)^6} \exp\left(-\frac{\log a}{(2x-1)^2}\right) \right|$$

$$= \frac{24a}{\log a} \times \frac{\log^2 a}{(2x-1)^4} \exp\left(-\frac{\log a}{(2x-1)^2}\right) + \frac{16a}{\log a} \times \frac{\log^3 a}{(2x-1)^6} \exp\left(-\frac{\log a}{(2x-1)^2}\right).$$

Taking $t = \frac{\log a}{(2x-1)^2} > 0$ and using $t^2 e^{-t} \leq 1$ and $t^3 e^{-t} \leq 2$,

$$|\Psi_a''(x)| \leq \frac{24a}{\log a} \times 1 + \frac{16a}{\log a} \times 2 \leq \frac{56e}{\log a}$$

since $a \leq e$. $\qquad\square$

**Lemma D.4.** *For all $x \in \mathbb{R}^T$, $\operatorname{supp}(\nabla F_{T,K,a}(x)) \subseteq \{1, \ldots, \operatorname{prog}^K(x) + 1\} \cup \operatorname{supp}(x)$, where $\operatorname{supp}(v) := \{\, i \in [d] : v_i \neq 0 \,\}$.*

*Proof.* Let $j = \operatorname{prog}^K(x)$ and $p = \operatorname{prog}^1(x)$, then

$$
\begin{aligned}
F_{T,K,a}(x) = &-\sum_{i=1}^{j+1} \Psi_a(x_{i-K}) \ldots \Psi_a(x_{i-2}) \Psi_a(x_{i-1}) \Phi(x_i) \\
&- \sum_{i=j+2}^{T} \Psi_a(x_{i-K}) \ldots \Psi_a(x_{i-2}) \Psi_a(x_{i-1}) \Phi(x_i) \\
&+ \sum_{i=1}^{p} \Gamma(x_i) + \sum_{i=p+1}^{T} \Gamma(x_i).
\end{aligned}
$$

Since $j = \operatorname{prog}^K(x)$, for all $i \geq j+2$, at least one of the values $x_{i-K}, \ldots, x_{i-2}, x_{i-1}$ is zero. Noting that $\Psi_a(0) = \Psi_a'(0) = 0$, the gradient of the second sum is zero. The first sum depends only on the first $j+1$ coordinates; thus, the gradient of the first sum is non-zero in at most the $(j+1)^{\text{th}}$ coordinate.

Since $p = \operatorname{prog}^1(x)$, the gradient of the last sum is zero because $\Gamma'(0) = 0$. Moreover, if $x_i = 0$, then $\Gamma'(x_i) = 0$; thus, $\nabla\left(\sum_{i=1}^{p} \Gamma(x_i)\right) \in \operatorname{supp}(x)$. $\qquad\square$

**Lemma D.5.** *For all $x \in \mathbb{R}^T$, if $\operatorname{prog}^K(x) < T$, then $\|\nabla F_{T,K,a}(x)\| > 1$.*

*Proof.* For all $j \in [T]$, the partial derivative of $F_{T,K,a}$ with respect to $x_j$ is

$$
\begin{aligned}
\frac{\partial F_{T,K,a}}{\partial x_j}(x) = \Big[ &- \Psi_a(x_{j-K}) \ldots \Psi_a(x_{j-1}) \Phi'(x_j) \\
&- \Psi_a(x_{j-K+1}) \ldots \Psi_a(x_{j-1}) \Psi_a'(x_j) \Phi(x_{j+1}) \\
&- \ldots \\
&- \Psi_a'(x_j) \Psi_a(x_{j+1}) \ldots \Psi_a(x_{\min\{j+K,T\}-1}) \Phi(x_{\min\{j+K,T\}}) \Big] + \Gamma'(x_j).
\end{aligned}
\tag{15}
$$

We now take the smallest $j \in [T]$ for which $x_j < 1$ and $x_{j-1} \geq 1, \ldots, x_{j-K} \geq 1$.

If such $j$ does not exists, then $x_1 \geq 1$ due to $x_0 = \cdots = x_{-K+1} \equiv 1$. Then $x_2 \geq 1$, and so on. Meaning that $x_j \geq 1$ for all $j \in [T]$, which contradicts the assumption of the theorem that $\operatorname{prog}^K(x) < T$.

Fixing such $j$, consider (15). There are two cases.
*Case 1: $x_j > -1$.* Note that $\Psi, \Phi, \Psi', \Phi' \geq 0$ are non-negative and $\Gamma' \leq 0$ is non-positive. Thus

$$\frac{\partial F_{T,K,a}}{\partial x_j}(x) \leq -\Psi_a(x_{j-K}) \ldots \Psi_a(x_{j-2}) \Psi_a(x_{j-1}) \Phi'(x_j).$$

Since $x_{j-1} \geq 1, \ldots, x_{j-K} \geq 1$ and $1 > x_j > -1$ (see Lemmas D.1 and D.3), we get

$$\frac{\partial F_{T,K,a}}{\partial x_j}(x) < -1.$$

*Case 2: $x_j \leq -1$.* Note that $\Psi, \Phi, \Psi', \Phi' \geq 0$ are non-negative. Thus

$$\frac{\partial F_{T,K,a}}{\partial x_j}(x) \leq \Gamma'(x_j).$$

Since $x_j \leq -1$ (see Lemma D.2), we get

$$\frac{\partial F_{T,K,a}}{\partial x_j}(x) < -1.$$

Finally, we can conclude that

$$\|\nabla F_{T,K,a}(x)\| \geq \left| \frac{\partial F_{T,K,a}}{\partial x_j}(x) \right| > 1.$$

$\square$

**Lemma D.6.** *Function $F_{T,K,a}$ satisfies*

$$F_{T,K,a}(0) - \inf_{x \in \mathbb{R}^T} F_{T,K,a}(x) \leq \Delta^0(K,a) \cdot T,$$

*where $\Delta^0(K,a) := \sqrt{2\pi e} \cdot a^K$.*

*Proof.* Since $\Gamma(0) = 0$ and $\Psi_a, \Phi \geq 0$, we get $F_{T,K,a}(0) \leq 0$. Next, due to $\Gamma(x) \geq 0, 0 \leq \Phi(x) \leq \sqrt{2\pi e}$ and $0 \leq \Psi_a(x) \leq a$ for all $x \in \mathbb{R}^d$,

$$F_{T,K,a}(x) \geq - \sum_{i=1}^{T} \Psi_a(x_{i-K}) \ldots \Psi_a(x_{i-2}) \Psi_a(x_{i-1}) \Phi(x_i) \geq -T\sqrt{2\pi e} \cdot a^K$$

for all $x \in \mathbb{R}^T$. $\square$

**Lemma D.7.** *For all $x \in \mathbb{R}^T$, $\|\nabla F_{T,K,a}(x)\|_{\infty} \leq \gamma_{\infty}(K,a)$, where $\gamma_{\infty}(K,a) := 6\sqrt{2\pi}e^{3/2} \cdot \frac{Ka^K}{\sqrt{\log a}}$.*

*Proof.* Using (15),

$$\left| \frac{\partial F_{T,K,a}}{\partial x_j}(x) \right| \leq \left| \Psi_a(x_{j-K}) \ldots \Psi_a(x_{j-1})\Phi'(x_j) \right.$$
$$+ \Psi_a(x_{j-K+1}) \ldots \Psi_a(x_{j-1})\Psi_a'(x_j)\Phi(x_{j+1})$$
$$+ \ldots$$
$$+ \left. \Psi_a'(x_j)\Psi_a(x_{j+1}) \ldots \Psi_a(x_{\min\{j+K,T\}-1})\Phi(x_{\min\{j+K,T\}}) \right| + |\Gamma'(x_j)|. \tag{16}$$

Thus,

$$\left| \frac{\partial F_{T,K,a}}{\partial x_j}(x) \right| \leq a^K\sqrt{e} + Ka^{K-1}\sqrt{2\pi e}\frac{2e}{\sqrt{\log a}} + e \leq 6\sqrt{2\pi}e^{3/2}\frac{Ka^K}{\sqrt{\log a}} \tag{17}$$

due to Lemmas D.1, D.2, and D.3. $\square$

**Lemma D.8.** *The function $F_{T,K,a}$ is $\ell_1(K,a)$–smooth, i.e., $\|\nabla^2 F_{T,K,a}(x)\| \leq \ell_1(K,a)$ for all $x \in \mathbb{R}^T$, where $\ell_1(K,a) := 12\sqrt{2\pi}e^{5/2} \cdot \frac{K^2a^K}{\log a}$.*

*Proof.* Taking the second partial derivative in (15),

$$\frac{\partial^2 F_{T,K,a}}{\partial x_j^2}(x) = \left[ - \Psi_a(x_{j-K}) \ldots \Psi_a(x_{j-1})\Phi''(x_j) \right.$$
$$- \Psi_a(x_{j-K+1}) \ldots \Psi_a(x_{j-1})\Psi_a''(x_j)\Phi(x_{j+1}) \tag{18}$$
$$- \ldots$$
$$- \left. \Psi_a''(x_j)\Psi_a(x_{j+1}) \ldots \Psi_a(x_{\min\{j+K,T\}-1})\Phi(x_{\min\{j+K,T\}}) \right] + \Gamma''(x_j).$$

Due to Lemmas D.1, D.2, and D.3,

$$\left| \frac{\partial^2 F_{T,K,a}}{\partial x_j^2}(x) \right| \leq \left[ 27a^K + K \times \frac{56\sqrt{2\pi}e^{3/2}a^{K-1}}{\log a} \right] + 27e^{-2} \leq 168\sqrt{2\pi}e^{3/2} \cdot \frac{Ka^K}{\log a} \tag{19}$$

Clearly, for all $\min\{j + K, T\} < i \leq T$,

$$\frac{\partial^2 F_{T,K,a}}{\partial x_j \partial x_i}(x) = 0 \tag{20}$$

due to the construction of $F_{T,K,a}$. Next, for all $j < i \leq \min\{j + K, T\}$,

$$
\begin{aligned}
\frac{\partial^2 F_{T,K,a}}{\partial x_j \partial x_i}(x) = \Big[ &- \Psi_a(x_{i-K}) \dots \Psi_a(x_{j-1}) \Psi_a'(x_j) \Psi_a(x_{j+1}) \dots \Psi_a(x_{i-1}) \Phi'(x_i) \\
&- \Psi_a(x_{i-K+1}) \dots \Psi_a(x_{j-1}) \Psi_a'(x_j) \Psi_a(x_{j+1}) \dots \Psi_a(x_{i-1}) \Psi_a'(x_i) \Phi(x_{i+1}) \\
&- \dots \\
&- \Psi_a'(x_j) \Psi_a(x_{j+1}) \dots \Psi_a(x_{i-1}) \Psi_a'(x_i) \Psi_a(x_{i+1}) \dots \Psi_a(x_{\min\{j+K,T\}-1}) \Phi(x_{\min\{j+K,T\}}) \Big],
\end{aligned}
$$

and

$$\left| \frac{\partial^2 F_{T,K,a}}{\partial x_j \partial x_i}(x) \right| \leq \frac{2e^{3/2} a^{K-1}}{\sqrt{\log a}} + (K - 1) \times \frac{4\sqrt{2\pi} e^{5/2} a^{K-2}}{\log a} \leq 4\sqrt{2\pi} e^{5/2} \cdot \frac{K a^K}{\log a} \tag{21}$$

for all $i \neq j \in [T]$ due to Lemmas D.1 and D.3 and $e \geq a > 1$

Notice that $\nabla^2 F_{T,K,a}$ is $(2K+1)$–diagonal Hessian. Repeating a textbook analysis for completeness and denoting temporary $\boldsymbol{H} := \nabla^2 F_{T,K,a}$, we will show that

$$\left\| \nabla^2 F_{T,K,a}(x) \right\| \leq (2K+1) \max_{i,j \in [T]} \left| \frac{\partial^2 F_{T,K,a}}{\partial x_j \partial x_i}(x) \right|. \tag{22}$$

for all $x \in \mathbb{R}^T$. Indeed, for all $x \in \mathbb{R}^T$ such that $\|x\| \leq 1$,

$$\left| x^\top \boldsymbol{H} x \right| = \left| \sum_{i=1}^T x_i \sum_{j=1}^T x_j \boldsymbol{H}_{ij} \right| = \left| \sum_{i=1}^T x_i \sum_{j=\max\{i-K,1\}}^{\min\{i+K,T\}} x_j \boldsymbol{H}_{ij} \right| \leq \max_{i,j \in [T]} |\boldsymbol{H}_{ij}| \left( \sum_{i=1}^T |x_i| \sum_{j=\max\{i-K,1\}}^{\min\{i+K,T\}} |x_j| \right),$$

where the second equality due to $\boldsymbol{H}$ is $(2K+1)$–diagonal. Using the Cauchy–Schwarz inequality,

$$\left| x^\top \boldsymbol{H} x \right| \leq \max_{i,j \in [T]} |\boldsymbol{H}_{ij}| \sqrt{\sum_{i=1}^T x_i^2 \sum_{i=1}^T \left( \sum_{j=\max\{i-K,1\}}^{\min\{i+K,T\}} |x_j| \right)^2} \leq \max_{i,j \in [T]} |\boldsymbol{H}_{ij}| \sqrt{\sum_{i=1}^T \left( \sum_{j=\max\{i-K,1\}}^{\min\{i+K,T\}} |x_j| \right)^2}$$

since $\|x\| \leq 1$. Next, using Jensen's inequality and $\|x\| \leq 1$,

$$
\begin{aligned}
\left| x^\top \boldsymbol{H} x \right| &\leq \max_{i,j \in [T]} |\boldsymbol{H}_{ij}| \sqrt{(2K+1) \sum_{i=1}^T \sum_{j=\max\{i-K,1\}}^{\min\{i+K,T\}} x_j^2} \leq \max_{i,j \in [T]} |\boldsymbol{H}_{ij}| \sqrt{(2K+1)^2 \sum_{i=1}^T x_i^2} \\
&\leq (2K+1) \max_{i,j \in [T]} |\boldsymbol{H}_{ij}|.
\end{aligned}
$$

We have proved (22). It is left to combine (22), (21), and (19).

$\square$

## E  PROOF OF THEOREM 4.2

**Theorem 4.2.** *Let $L, \Delta, \varepsilon, n, \sigma^2, d, \tau_s, \tau_w, h > 0$ be any numbers such that $\bar{c}_1 \varepsilon \log^4(n+1) < L\Delta$ and dimension $d \geq \bar{c}_3 \frac{L\Delta}{\log^3(n+1)\varepsilon}$. Consider Protocol 1. For all $i \in [n]$ and $k \geq 0$, compressor $\mathcal{C}_i^k$ selects and transmits $P_i^k$ uniformly random coordinates without replacement, scaled by any constants[7], where $P_i^k \in \{0, \dots, d\}$ may vary across each compressor [8]. Then, for any algorithm $A \in \mathcal{A}_{zr}$ (Def. 2.3),*

---

[7]To potentially preserve unbiasedness. For instance, Rand$K$ scales by $d/K$.

[8]For instance, the compressors can be Rand$K$ (see Def. C.1) with any $K \in [d]$, Perm$K$ (Szlendak et al., 2021), Identity compressor when $P_i^k = d$.

*there exists a function $f : \mathbb{R}^d \to \mathbb{R}$ such that $f$ is $L$-smooth, i.e., $\|\nabla f(x) - \nabla f(y)\| \leq L \|x - y\|$ for all $x, y \in \mathbb{R}^d$, and $f(0) - \inf_{x \in \mathbb{R}^d} f(x) \leq \Delta$, exists a stochastic gradient oracles that satisfies Assumption 1.3, and $\mathbb{E}\left[\inf_{y \in S_t} \|\nabla f(y)\|^2\right] > \varepsilon$ for all*

$$t \leq \bar{c}_2 \times \left( \frac{1}{\log^3(n+1)} \cdot \frac{L\Delta}{\varepsilon} \right) \min \left\{ \frac{1}{\log(n+1)} \cdot \tau_s d, \max \left\{ h, \frac{1}{\log^3(n+1)} \cdot \frac{h\sigma^2}{\varepsilon} \right\} \right\}, \qquad (12)$$

*where $S_t$ is the set of all possible points that can be constructed by $A$ up to time $t$ based on $I$ and $\{I_i\}$. The quantities $\bar{c}_1, \bar{c}_2$, and $\bar{c}_3$ are universal constants.*

*Proof.* **(Step 1: Construction).** Using the construction from Section D.1, we define a scaled version of it. Let us take any $\lambda > 0$, $d, T \in \mathbb{N}$, $d \geq T$, and take the function $f : \mathbb{R}^d \to \mathbb{R}$ such that

$$f(x) := \frac{L\lambda^2}{\ell_1(K, a)} F_{T,K,a} \left( \frac{x_{[T]}}{\lambda} \right),$$

where $\ell_1(K, a)$ is defined in Lemma D.8 and $x_{[T]} \in \mathbb{R}^T$ is the vector with the first $T$ coordinates of $x \in \mathbb{R}^d$. Notice that the last $d - T$ coordinates are artificial.

First, we have to show that $f$ is $L$-smooth and $f(0) - \inf_{x \in \mathbb{R}^d} f(x) \leq \Delta$, Using Lemma D.8,

$$\|\nabla f(x) - \nabla f(y)\| = \frac{L\lambda}{\ell_1(K, a)} \left\| \nabla F_{T,K,a} \left( \frac{x_{[T]}}{\lambda} \right) - \nabla F_{T,K,a} \left( \frac{y_{[T]}}{\lambda} \right) \right\| \leq L\lambda \left\| \frac{x_{[T]}}{\lambda} - \frac{y_{[T]}}{\lambda} \right\|$$

$$= L \left\| x_{[T]} - y_{[T]} \right\| \leq L \left\| x - y \right\| \quad \forall x, y \in \mathbb{R}^d.$$

Let us take

$$T = \left\lfloor \frac{\Delta \cdot \ell_1(K, a)}{L\lambda^2 \cdot \Delta^0(K, a)} \right\rfloor.$$

Due to Lemma D.6,

$$f(0) - \inf_{x \in \mathbb{R}^d} f(x) = \frac{L\lambda^2}{\ell_1(K, a)} (F_{T,K,a}(0) - \inf_{x \in \mathbb{R}^T} F_{T,K,a}(x)) \leq \frac{L\lambda^2 \Delta^0(K, a) T}{\ell_1(K, a)} \leq \Delta,$$

where $\Delta^0(K, a)$ is defined in Lemma D.6. We also choose

$$\lambda = \frac{\sqrt{2\varepsilon} \ell_1(K, a)}{L} \qquad (23)$$

to ensure that

$$\|\nabla f(x)\|^2 = \frac{L^2\lambda^2}{\ell_1^2(K, a)} \left\| \nabla F_{T,K,a} \left( \frac{x_{[T]}}{\lambda} \right) \right\|^2 = 2\varepsilon \left\| \nabla F_{T,K,a} \left( \frac{x_{[T]}}{\lambda} \right) \right\|^2 > 2\varepsilon \cdot \mathbb{1} \left[ \text{prog}^K(x_{[T]}) < T \right],$$

$$(24)$$

where the last inequality due to Lemma D.5. Note that

$$T = \left\lfloor \frac{L\Delta}{2\Delta^0(K, a) \cdot \ell_1(K, a) \cdot \varepsilon} \right\rfloor. \qquad (25)$$

**(Step 2: Stochastic Oracle).**
We take the stochastic oracle construction form (Arjevani et al., 2022). For all $j \in [d]$,

$$[\nabla f(x; \xi)]_j := \nabla_j f(x) \left( 1 + \mathbb{1} \left[ j = \text{prog}^K(x) + 1 \right] \left( \frac{\xi}{p_\sigma} - 1 \right) \right) \quad \forall x \in \mathbb{R}^d, \qquad (26)$$

and $\mathcal{D}_\xi = \text{Bernouilli}(p_\sigma)$, where $p_\sigma \in (0, 1]$. We denote $[x]_j$ as the $j^{\text{th}}$ index of a vector $x \in \mathbb{R}^d$. It is left to show this mapping is unbiased and $\sigma^2$-variance-bounded. Indeed,

$$\mathbb{E}[[\nabla f(x, \xi)]_i] = \nabla_i f(x) \left( 1 + \mathbb{1} \left[ i = \text{prog}^K(x) + 1 \right] \left( \frac{\mathbb{E}[\xi]}{p_\sigma} - 1 \right) \right) = \nabla_i f(x)$$

for all $i \in [d]$, and

$$\mathbb{E}\left[ \|\nabla f(x; \xi) - \nabla f(x)\|^2 \right] \leq \max_{j \in [d]} |\nabla_j f(x)|^2 \mathbb{E}\left[ \left( \frac{\xi}{p_\sigma} - 1 \right)^2 \right]$$

because the difference is non-zero only in one coordinate. Thus

$$\mathbb{E}\left[\|\nabla f(x,\xi) - \nabla f(x)\|^2\right] \leq \frac{\|\nabla f(x)\|_\infty^2 (1-p_\sigma)}{p_\sigma} = \frac{L^2\lambda^2 \left\|F_{T,K,a}\left(\frac{x_{[T]}}{\lambda}\right)\right\|_\infty^2 (1-p_\sigma)}{\ell_1^2(K,a)p_\sigma}$$

$$\leq \frac{L^2\lambda^2\gamma_\infty^2(K,a)(1-p_\sigma)}{\ell_1^2(K,a)p_\sigma},$$

where we use Lemma D.7 in the last inequality. Taking

$$p_\sigma = \min\left\{\frac{L^2\lambda^2\gamma_\infty^2(K,a)}{\sigma^2\ell_1^2(K,a)}, 1\right\} \overset{(23)}{=} \min\left\{\frac{2\varepsilon\gamma_\infty^2(K,a)}{\sigma^2}, 1\right\}, \tag{27}$$

we ensure that $\mathbb{E}\left[\|\nabla f(x,\xi) - \nabla f(x)\|^2\right] \leq \sigma^2$.

**(Step 3: Reduction to the Analysis of Concentration).** At the beginning, due to Definition 2.3, $s_i^0 = 0$ for all $i \in [n]$. Thus, if $k = 0$, all workers can receive zero vectors from the server. Thus, at the beginning, all workers can only calculate stochastic gradients at the point zero.

Let $t^{1,i}$ denote the earliest time at which all of the first $K$ coordinates become non-zero in the local information available to worker $i$. In other words, $t^{1,i}$ is the first time when worker $i$ has *discovered* all of the first $K$ coordinates. Consequently, prior to time

$$t_1 := \min_{i \in [n]} t_{1,i} \tag{28}$$

neither the server nor any worker is able to discover (filled with non-zero values) the $(K+1)^{\text{th}}$ and subsequent coordinates due to Lemma D.4.

**There are two options by which a worker may discover a new non-zero coordinate: through local stochastic computations or through communication from the server.**

**Option 1:** In the first option, a worker computes a stochastic gradient, which takes $h$ seconds. However, due to the construction of stochastic gradients, even if the computation is completed, the worker will not make progress or discover a new non-zero coordinate, as it will be zeroed out with probability $p_\sigma$. Due to Lemma D.4, each worker can discover at most one coordinate at position $\text{prog}^K(x) + 1$ before time $t_1$ in the first $K$ coordinates, where $x$ is a query point.

*Remark* E.1. For this reason, making multiple queries with the same random variable instead of a single query does not help the algorithm progress: if the coordinates are zeroed out, then they are zeroed out in all vectors.

Let $\eta_{1,i,1}$ be the number of stochastic gradients computations[9] until the first moment when a coordinate is not zeroed out in (26) in worker $i$. Assume that $\xi_1, \xi_2, \ldots$ is a stream of i.i.d. random Bernoulli variables from (26) in worker $i$ (all workers have different streams), then

$$\mathbb{P}\left(\eta_{1,i,1} \leq t\right) \leq \sum_{k=1}^{\lfloor t \rfloor} \mathbb{P}\left(\xi_k = 1, \xi_{k-1} = 0, \ldots, \xi_1 = 0\right) = \sum_{k=1}^{\lfloor t \rfloor} p_\sigma(1-p_\sigma)^{j-1} \leq tp_\sigma.$$

for all $t \geq 0$. Similarly, let $\eta_{1,i,k}$ denote the number of stochastic gradient computations until the first moment when a coordinate is not zeroed out in (26), after the moment when the $(k-1)^{\text{th}}$ coordinate is no longer zeroed out in worker $i$. In other words, worker $i$ should calculate $\eta_{1,i,1}$ stochastic gradients to discover the first coordinate, calculate $\eta_{1,i,2}$ stochastic gradients to discover the second coordinate, and so on. Since the draws of $\xi$ in (26) are i.i.d., we can conclude that

$$\mathbb{P}\left(\eta_{1,i,k} \leq t | \eta_{1,i,k-1}, \ldots, \eta_{1,i,1}\right) \leq tp_\sigma$$

for all $k \geq 1$ and $t \geq 0$.

**Option 2:** In the second option, worker $i$ receives $P \in \{0, \ldots, d\}$ random coordinates with the set of indices $\{\nu_{1,1}, \ldots, \nu_{1,P}\}$ *without replacement*, where it takes $\tau_s$ seconds to receive one

---

[9]It is possible that $\mathbb{P}\left(\eta_{1,i,1} = \infty\right) > 0$ if, for instance, the algorithm decides to stop calculating stochastic gradients. And even $\mathbb{P}\left(\eta_{1,i,1} = \infty\right) = 1$ if it does not calculate at all.

coordinate. Then, the worker receives $\bar{P} \in \{0, \ldots, d\}$ random coordinates with the set of indices $\{\nu_{2,1}, \ldots, \nu_{2,\bar{P}}\}$ *without replacement*, and so on (all workers get different sets; we drop the indices of the workers in the notations).

Consequently, the worker receives a stream of coordinate indices $(\nu_1, \nu_2, \ldots)$ where we concatenated the sets of indices, preserving the exact order in which the server sampled them. Note that all workers have different streams, and we now focus on one worker.

Notice that

$$\mathbb{P}(\nu_1 \in [K]) = \frac{K}{d}$$

since $\nu_1$ is uniformly random coordinate from the set $[d]$. Next,

$$\mathbb{P}(\nu_2 \in [K] | \nu_1) \leq \frac{K}{d-1},$$

because either $\nu_1 \in [K]$, in which case the probability is $\frac{K-1}{d-1}$, or $\nu_1 \notin [K]$, in which case the probability is $\frac{K}{d-1}$. Using the same reasoning,

$$\mathbb{P}(\nu_i \in [K] | \nu_1, \ldots, \nu_{i-1}) \leq \frac{K}{d-i+1}.$$

for all $i \leq d$.

Hence, the worker receives a stream of coordinates $\nu_1, \nu_2, \ldots$ such that $\mathbb{P}(\nu_i \in [K] | \nu_1, \ldots, \nu_{i-1}) \leq \frac{K}{d-i+1}$ for all $i \leq d$. Let $\mu_{1,i,1}$ be the number of received coordinates until the moment when the last received coordinate belongs to $[K]$ in worker $i$. Similarly, let $\mu_{1,i,k}$ be the number of received coordinates until the moment when the last received coordinate belongs to $[K]$, after the $(k-1)^{\text{th}}$ time this has happened in worker $i$. In other words, worker $i$ should receive $\mu_{1,i,1}$ coordinates to obtain a coordinate that belongs to $[K]$. To get the next coordinate that belongs to $[K]$, the worker should receive $\mu_{1,i,2}$ coordinates, and so on. Then,

$$\mathbb{P}(\mu_{1,i,1} = j) = \mathbb{P}(\nu_j \in [K], \nu_{j-1} \notin [K], \ldots, \nu_1 \notin [K])$$

$$= \mathbb{P}(\nu_j \in [K] | \nu_{j-1} \notin [K], \ldots, \nu_1 \notin [K]) \mathbb{P}(\nu_{j-1} \notin [K], \ldots, \nu_1 \notin [K]) \leq \frac{K}{d-j+1}$$

for all $1 \leq j \leq d$, and

$$\mathbb{P}(\mu_{1,i,1} \leq t) = \sum_{j=1}^{\lfloor t \rfloor} \mathbb{P}(\mu_{1,i,1} = j) \leq \frac{Kt}{d-t+1}. \tag{29}$$

for all $0 \leq t \leq d$. Similarly,

$$\mathbb{P}(\mu_{1,i,k} = j | \mu_{1,i,k-1}, \ldots, \mu_{1,i,1})$$
$$= \mathbb{P}(\nu_{u+j} \in [K], \nu_{u+j-1} \notin [K], \ldots, \nu_{u+1} \notin [K] | \nu_u \in [K], \ldots, \nu_1 \notin [K])$$
$$\leq \frac{K}{\max\{d-u, 0\} - j + 1},$$

where $u = \sum_{j=1}^{k-1} \mu_{1,i,j}$, for all $j \leq \max\{d-u, 0\}$. Thus,

$$\mathbb{P}(\mu_{1,i,k} \leq t | \mu_{1,i,k-1}, \ldots, \mu_{1,i,1}) \leq \frac{Kt}{\max\{d - \sum_{j=1}^{k-1} \mu_{1,i,j}, 0\} - t + 1},$$

for all $0 \leq t \leq \max\{d - \sum_{j=1}^{k-1} \mu_{1,i,j}, 0\}$.

Recall that the workers can discover new non-zero coordinates only through the stochastic processes discussed above. To discover all of the first $K$ coordinates, either the first or the second process must uncover at least $\frac{K}{2}$ coordinates[10]. If worker $i$ has discovered fewer than $\frac{K}{2}$ coordinates through

---

[10]At the end of the proof, we take $K \bmod 2 = 0$.

stochastic gradient computations and fewer than $\frac{K}{2}$ coordinates through receiving coordinates from the server, then it will not be able to cover all $K$ coordinates. Hence,

$$t_1 \geq \min_{i \in [n]} \left\{ \min \left\{ h \sum_{k=1}^{\frac{K}{2}} \eta_{1,i,k}, \tau_{\mathrm{s}} \sum_{k=1}^{\frac{K}{2}} \mu_{1,i,k} \right\} \right\}, \tag{30}$$

where $t_1$ is defined in (28). This is because $h \sum_{k=1}^{\frac{K}{2}} \eta_{1,i,k}$ is the time required to obtain $\frac{K}{2}$ "lucky" stochastic gradients, those for which the coordinates are not zeroed out, and $\tau_{\mathrm{s}} \sum_{k=1}^{\frac{K}{2}} \mu_{1,i,k}$ is the time required to receive $\frac{K}{2}$ "lucky" coordinates that belong to $[K]$.

*Remark* E.2. The previous derivations hold for all $\tau_{\mathrm{w}} > 0$. If we start taking the communication time $\tau_{\mathrm{w}}$ into account, then the bound on $t_1$ in (30) may only increase. For all $\tau_{\mathrm{w}} > 0$, worker $i$ still has to discover new non-zero coordinates either through stochastic gradient computations or by receiving coordinates from the server and it will take at least

$$\min_{i \in [n]} \left\{ \min \left\{ h \sum_{k=1}^{\frac{K}{2}} \eta_{1,i,k}, \tau_{\mathrm{s}} \sum_{k=1}^{\frac{K}{2}} \mu_{1,i,k} \right\} \right\}$$

seconds to discover all of the first $K$ coordinates.

Once the workers have discovered the first $K$ coordinates, the discovery process repeats for the set $\{K+1, \ldots, 2K\}$, which similarly requires at least

$$\min_{i \in [n]} \left\{ \min \left\{ h \sum_{k=1}^{\frac{K}{2}} \eta_{2,i,k}, \tau_{\mathrm{s}} \sum_{k=1}^{\frac{K}{2}} \mu_{2,i,k} \right\} \right\}$$

seconds, where $\{\eta_{b,i,k}\}$ and $\{\mu_{b,i,k}\}$ are random variables such that

$$\mathbb{P}\left( \eta_{b,i,k} \leq t | \eta_{b,i,k-1}, \ldots, \eta_{b,i,1}, \mathcal{G}_{b-1} \right) \leq t p_\sigma \tag{31}$$

for all $b \geq 1, k \geq 1, i \in [n], t \geq 0$, and

$$\mathbb{P}\left( \mu_{b,i,k} \leq t | \mu_{b,i,k-1}, \ldots, \mu_{b,i,1}, \mathcal{G}_{b-1} \right) \leq \frac{Kt}{\max\{d - \sum_{j=1}^{k-1} \mu_{b,i,j}, 0\} - t + 1} \tag{32}$$

for all $b \geq 1, k \geq 1, i \in [n]$, and $t \leq \max\{d - \sum_{j=1}^{k-1} \mu_{b,i,j}, 0\}$, where $\mathcal{G}_{b-1}$ is the sigma-algebra generated by $\{\eta_{b',i,k}\}_{i \in [n], k \in [\frac{K}{2}], b' < b}$ and $\{\mu_{b',i,k}\}_{i \in [n], k \in [\frac{K}{2}], b' < b}$ and $u = \sum_{j=1}^{k-1} \mu_{b,i,j}$.

More formally, $\eta_{2,i,k}$ can be defined as the number of stochastic gradient computations until the first moment when a coordinate is not zeroed out in (26), after the moment when the $(k-1)^{\mathrm{th}}$ coordinate is no longer zeroed out, when $\mathrm{prog}^K$ of the input points to the stochastic gradients is $\geq K$, and $\mu_{2,i,k}$ be the number of received coordinates until the moment when the last received coordinate belongs to $\{K+1, \ldots, 2K\}$, after the $(k-1)^{\mathrm{th}}$ time this has happened, when $\mathrm{prog}^1$ of the input points to the compressor is $\geq K+1$, and so on.

We define

$$p_K := \frac{2K}{d}. \tag{33}$$

Finally, to discover the $T^{\mathrm{th}}$ coordinates it takes at least

$$\sum_{b=1}^{B} \min_{i \in [n]} \left\{ \min \left\{ h \sum_{k=1}^{\frac{K}{2}} \eta_{b,i,k}, \tau_{\mathrm{s}} \sum_{k=1}^{\frac{K}{2}} \mu_{b,i,k} \right\} \right\}$$

seconds, where $B = \left\lfloor \frac{T}{K} \right\rfloor$. It it left to use the following lemma.

**Lemma E.3.** *Let $\{\eta_{b,i,j}\}_{i,j,b\geq 0}$ and $\{\mu_{b,i,j}\}_{i,j,b\geq 0}$ be random variables such that*

$$\mathbb{P}\left(\eta_{b,i,k}\leq t|\eta_{b,i,k-1},\ldots,\eta_{b,i,1},\mathcal{G}_{b-1}\right)\leq tp_\sigma \tag{34}$$

*for all $b\geq 1, k\geq 1, i\in[n], t\geq 0$, and*

$$\mathbb{P}\left(\mu_{b,i,k}\leq t|\mu_{b,i,k-1},\ldots,\mu_{b,i,1},\mathcal{G}_{b-1}\right)\leq\frac{Kt}{\max\{d-\sum_{j=1}^{k-1}\mu_{b,i,j},0\}-t+1}, \tag{35}$$

*for all $b\geq 1, k\geq 1, i\in[n], 0\leq t\leq\max\{d-\sum_{j=1}^{k-1}\mu_{b,i,j},0\}$, and $1\leq K\leq d$, where $\mathcal{G}_{b-1}$ is the sigma-algebra generated by $\{\eta_{b',i,k}\}_{i\in[n],k\in[\frac{K}{2}],b'<b}$ and $\{\mu_{b',i,k}\}_{i\in[n],k\in[\frac{K}{2}],b'<b}$. Then*

$$\mathbb{P}\left(\sum_{b=1}^B\min_{i\in[n]}\left\{\min\left\{h\sum_{k=1}^{\frac{K}{2}}\eta_{b,i,k},\tau_s\sum_{k=1}^{\frac{K}{2}}\mu_{b,i,k}\right\}\right\}\leq\bar{t}\right)\leq\delta$$

*with*

$$\bar{t}:=\frac{BK+\log\delta}{e^4(2n)^{2/K}(4+\frac{2}{K}\log(2n))}\min\left\{\frac{h}{p_\sigma},\frac{\tau_s}{p_K}\right\}, \tag{36}$$

*where*

$$p_K:=\frac{2K}{d}.$$

**(Step 4: Endgame).** Thus, with probability at least $1-\delta$, any zero-respecting algorithm requires at least $\bar{t}$ seconds to discover the $T^{\text{th}}$ coordinate. Since $\text{prog}^K(x)\leq\text{prog}^1(x)$ for all $x\in\mathbb{R}^T$, and due to (24),

$$\inf_{y\in S_t}\|\nabla f(y)\|^2>2\varepsilon\inf_{y\in S_t}\mathbb{1}\left[\text{prog}^1(y_{[T]})<T\right],$$

where $S_t$ is the set of all possible candidate points to be an $\varepsilon$–stationary point up to time $t$, which can be computed by $A$. Taking $\delta=\frac{1}{2}$,

$$\mathbb{E}\left[\inf_{y\in S_t}\|\nabla f(y)\|^2\right]>2\varepsilon\mathbb{E}\left[\inf_{y\in S_t}\mathbb{1}\left[\text{prog}^1(y_{[T]})<T\right]\right]\geq\varepsilon,$$

for $t=\frac{1}{2}\bar{t}$ because $\text{prog}^1(y_{[T]})<T$ for all $y\in S_t$ with probability at least $\frac{1}{2}$.

It is left to choose $K$ and $a$, and substitute all quantities to $\bar{t}$. Using $B=\lfloor\frac{T}{K}\rfloor$,

$$\bar{t}=\frac{\lfloor\frac{T}{K}\rfloor K-\log 8}{e^4(2n)^{2/K}(4+\frac{2}{K}\log(2n))}\min\left\{\frac{h}{p_\sigma},\frac{\tau_s}{p_K}\right\}$$

$$\geq\frac{T-K-\log 8}{e^4(2n)^{2/K}(4+\frac{2}{K}\log(2n))}\min\left\{\frac{h}{p_\sigma},\frac{\tau_s}{p_K}\right\}.$$

Due to (25), (27), and (33),

$$\bar{t}\geq\frac{\left\lfloor\frac{L\Delta}{2\Delta^0(K,a)\cdot\ell_1(K,a)\cdot\varepsilon}\right\rfloor-K-\log 8}{e^4(2n)^{2/K}(4+\frac{2}{K}\log(2n))}\min\left\{\max\left\{h,\frac{h\sigma^2}{2\varepsilon\gamma_\infty^2(K,a)}\right\},\frac{\tau_s d}{2K}\right\}.$$

Using the definitions of $\Delta^0(K,a)$, $\gamma_\infty(K,a)$, and $\ell_1(K,a)$,

$$\bar{t}\geq\left(e^4(2n)^{2/K}\left(4+\frac{2}{K}\log(2n)\right)\right)^{-1}\left(\left\lfloor\frac{L\Delta\log a}{48\pi e^3 K^2 a^{2K}\varepsilon}\right\rfloor-K-\log 8\right)\min\left\{\max\left\{h,\frac{h\sigma^2\log a}{144\pi e^3 K^2 a^{2K}\varepsilon}\right\},\frac{\tau_s d}{2K}\right\}.$$

We can take any $a$ from the interval $(1,e]$. We choose $a=1+\frac{1}{K}$, then $\log a=\log\left(1+\frac{1}{K}\right)\geq\frac{1}{2K}$ for all $K\geq 1$, $a^{2K}\leq e^2$ for all $K\geq 1$, and

$$\bar{t}\geq\left(e^4(2n)^{2/K}\left(4+\frac{2}{K}\log(2n)\right)\right)^{-1}\left(\left\lfloor\frac{L\Delta}{96\pi e^5 K^3\varepsilon}\right\rfloor-K-\log 8\right)\min\left\{\max\left\{h,\frac{h\sigma^2}{288\pi e^5 K^3\varepsilon}\right\},\frac{\tau_s d}{2K}\right\}.$$

Taking $K = 2\lceil 2\log(2n)\rceil$, $(2n)^{2/K} \le e$, $K \le 16\log(n+1)$ and

$$\bar{t} \ge \frac{1}{5e^5}\left(\left\lfloor\frac{L\Delta}{96\cdot 8^4\pi e^5\log^3(n+1)\varepsilon}\right\rfloor - 32\log(n+1)\right)\min\left\{\max\left\{h, \frac{h\sigma^2}{288\cdot 8^4\pi e^5\log^3(n+1)\varepsilon}\right\}, \frac{\tau_{\rm s}d}{32\log(n+1)}\right\}$$

$$\ge \frac{1}{5e^5}\left(\frac{L\Delta}{96\cdot 8^4\pi e^5\log^3(n+1)\varepsilon} - 36\log(n+1)\right)\min\left\{\max\left\{h, \frac{h\sigma^2}{288\cdot 8^4\pi e^5\log^3(n+1)\varepsilon}\right\}, \frac{\tau_{\rm s}d}{32\log(n+1)}\right\}.$$

We assume $\frac{L\Delta}{\varepsilon} \ge \bar{c}_1\log^4(n+1)$ for a universal constant $\bar{c}_1$. Taking $\bar{c}_1$ large enough, one can see that

$$\bar{t} \ge \frac{1}{5e^5}\left(\frac{L\Delta}{2\cdot 96\cdot 8^4\pi e^5\log^3(n+1)\varepsilon}\right)\min\left\{\max\left\{h, \frac{h\sigma^2}{288\cdot 8^4\pi e^5\log^3(n+1)\varepsilon}\right\}, \frac{\tau_{\rm s}d}{32\log(n+1)}\right\}.$$

For a small enough universal $\bar{c}_2$, we get the inequality

$$\bar{t} \ge \bar{c}_2 \times \left(\frac{L\Delta}{\log^3(n+1)\varepsilon}\right)\min\left\{\max\left\{h, \frac{h\sigma^2}{\log^3(n+1)\varepsilon}\right\}, \frac{\tau_{\rm s}d}{\log(n+1)}\right\},$$

which finishes the proof. Notice that we can take

$$d \ge T = \left\lfloor\frac{L\Delta\log a}{48\pi e^3K^2a^{2K}\varepsilon}\right\rfloor = \Theta\left(\frac{L\Delta}{\log^3(n+1)\varepsilon}\right).$$

$\square$

### E.1 MAIN CONCENTRATION LEMMA

**Lemma E.3.** *Let $\{\eta_{b,i,j}\}_{i,j,b\ge 0}$ and $\{\mu_{b,i,j}\}_{i,j,b\ge 0}$ be random variables such that*

$$\mathbb{P}\left(\eta_{b,i,k} \le t | \eta_{b,i,k-1},\dots,\eta_{b,i,1},\mathcal{G}_{b-1}\right) \le tp_\sigma \tag{34}$$

*for all $b \ge 1, k \ge 1, i \in [n], t \ge 0$, and*

$$\mathbb{P}\left(\mu_{b,i,k} \le t | \mu_{b,i,k-1},\dots,\mu_{b,i,1},\mathcal{G}_{b-1}\right) \le \frac{Kt}{\max\{d - \sum_{j=1}^{k-1}\mu_{b,i,j}, 0\} - t + 1}, \tag{35}$$

*for all $b \ge 1, k \ge 1, i \in [n], 0 \le t \le \max\{d - \sum_{j=1}^{k-1}\mu_{b,i,j}, 0\}$, and $1 \le K \le d$, where $\mathcal{G}_{b-1}$ is the sigma-algebra generated by $\{\eta_{b',i,k}\}_{i\in[n],k\in[\frac{K}{2}],b'<b}$ and $\{\mu_{b',i,k}\}_{i\in[n],k\in[\frac{K}{2}],b'<b}$. Then*

$$\mathbb{P}\left(\sum_{b=1}^{B}\min_{i\in[n]}\left\{\min\left\{h\sum_{k=1}^{\frac{K}{2}}\eta_{b,i,k}, \tau_{\rm s}\sum_{k=1}^{\frac{K}{2}}\mu_{b,i,k}\right\}\right\} \le \bar{t}\right) \le \delta$$

*with*

$$\bar{t} := \frac{BK + \log\delta}{e^4(2n)^{2/K}(4 + \frac{2}{K}\log(2n))}\min\left\{\frac{h}{p_\sigma}, \frac{\tau_{\rm s}}{p_K}\right\}, \tag{36}$$

*where*

$$p_K := \frac{2K}{d}.$$

*Proof.* Let us temporarily define $\beta_{b,i} := \min\left\{h\sum_{k=1}^{\frac{K}{2}}\eta_{b,i,k}, \tau_{\rm s}\sum_{k=1}^{\frac{K}{2}}\mu_{b,i,k}\right\}$. Using Chernoff's method, we get

$$\mathbb{P}\left(\sum_{b=1}^{B}\min_{i\in[n]}\beta_{b,i} \le \bar{t}\right) = \mathbb{P}\left(\exp\left(-\sum_{b=1}^{B}\lambda\min_{i\in[n]}\beta_{b,i}\right) \ge \exp\left(-\lambda\bar{t}\right)\right)$$

$$\le \exp\left(\lambda\bar{t}\right)\mathbb{E}\left[\exp\left(-\sum_{b=1}^{B}\lambda\min_{i\in[n]}\beta_{b,i}\right)\right] \tag{37}$$

$$= \exp\left(\lambda\bar{t}\right)\mathbb{E}\left[\mathbb{E}\left[\exp\left(-\lambda\min_{i\in[n]}\beta_{B,i}\right)\Big|\mathcal{G}_{B-1}\right]\exp\left(-\sum_{b=1}^{B-1}\lambda\min_{i\in[n]}\beta_{b,i}\right)\right]$$

for all $\lambda > 0$ since $\beta_{1,i}, \ldots, \beta_{B-1,i}$ are $\mathcal{G}_{B-1}$–measurable. Consider the inner expectation separately:

$$\mathbb{E}\left[\exp\left(-\lambda \min_{i\in[n]} \beta_{B,i}\right)\Big|\mathcal{G}_{B-1}\right] = \mathbb{E}\left[\max_{i\in[n]} \exp\left(-\lambda\beta_{B,i}\right)\Big|\mathcal{G}_{B-1}\right] \le \sum_{i=1}^{n} \mathbb{E}\left[\exp\left(-\lambda\beta_{B,i}\right)|\mathcal{G}_{B-1}\right],$$

where we bound $\max$ by $\sum$. Using the temporal definitions of $\{\beta_{B,i}\}$,

$$\begin{aligned}
&\mathbb{E}\left[\exp\left(-\lambda \min_{i\in[n]} \beta_{B,i}\right)\Big|\mathcal{G}_{B-1}\right] \\
&\le \sum_{i=1}^{n} \mathbb{E}\left[\exp\left(-\lambda \min\left\{h\sum_{k=1}^{\frac{K}{2}}\eta_{B,i,k}, \tau_{\mathsf{s}}\sum_{k=1}^{\frac{K}{2}}\mu_{B,i,k}\right\}\right)\Big|\mathcal{G}_{B-1}\right] \\
&= \sum_{i=1}^{n} \mathbb{E}\left[\max\left\{\exp\left(-\lambda h\sum_{k=1}^{\frac{K}{2}}\eta_{B,i,k}\right), \exp\left(-\lambda\tau_{\mathsf{s}}\sum_{k=1}^{\frac{K}{2}}\mu_{B,i,k}\right)\right\}\Big|\mathcal{G}_{B-1}\right] \\
&\le \underbrace{\sum_{i=1}^{n}\mathbb{E}\left[\exp\left(-\lambda h\sum_{k=1}^{\frac{K}{2}}\eta_{B,i,k}\right)\Big|\mathcal{G}_{B-1}\right]}_{I_1:=} + \underbrace{\sum_{i=1}^{n}\mathbb{E}\left[\exp\left(-\lambda\tau_{\mathsf{s}}\sum_{k=1}^{\frac{K}{2}}\mu_{B,i,k}\right)\Big|\mathcal{G}_{B-1}\right]}_{I_2^{\frac{K}{2}}:=}.
\end{aligned} \tag{38}$$

Using the tower property,

$$I_1 = \sum_{i=1}^{n}\mathbb{E}\left[\underbrace{\mathbb{E}\left[\exp\left(-\lambda h\eta_{B,i,\frac{K}{2}}\right)\Big|\eta_{B,i,\frac{K}{2}-1},\ldots,\eta_{B,i,1},\mathcal{G}_{B-1}\right]}_{J_1:=}\exp\left(-\lambda h\sum_{k=1}^{\frac{K}{2}-1}\eta_{B,i,k}\right)\Big|\mathcal{G}_{B-1}\right]. \tag{39}$$

Next,

$$\begin{aligned}
J_1 &:= \mathbb{E}\left[\exp\left(-\lambda h\eta_{B,i,\frac{K}{2}}\right)\Big|\eta_{B,i,\frac{K}{2}-1},\ldots,\eta_{B,i,1},\mathcal{G}_{B-1}\right] \\
&\le \exp\left(-\lambda t\right)\mathbb{P}\left(h\eta_{B,i,\frac{K}{2}} > t\Big|\eta_{B,i,\frac{K}{2}-1},\ldots,\eta_{B,i,1},\mathcal{G}_{B-1}\right) \\
&\quad + \mathbb{P}\left(h\eta_{B,i,\frac{K}{2}} \le t\Big|\eta_{B,i,\frac{K}{2}-1},\ldots,\eta_{B,i,1},\mathcal{G}_{B-1}\right) \\
&= \exp\left(-\lambda t\right) + (1-\exp\left(-\lambda ht\right))\mathbb{P}\left(h\eta_{B,i,\frac{K}{2}} \le t\Big|\eta_{B,i,\frac{K}{2}-1},\ldots,\eta_{B,i,1},\mathcal{G}_{B-1}\right) \\
&\le \exp\left(-\lambda t\right) + \mathbb{P}\left(h\eta_{B,i,\frac{K}{2}} \le t\Big|\eta_{B,i,\frac{K}{2}-1},\ldots,\eta_{B,i,1},\mathcal{G}_{B-1}\right) \\
&= \exp\left(-\lambda t\right) + \mathbb{P}\left(\eta_{B,i,\frac{K}{2}} \le \frac{t}{h}\Big|\eta_{B,i,\frac{K}{2}-1},\ldots,\eta_{B,i,1},\mathcal{G}_{B-1}\right)
\end{aligned}$$

for all $t \ge 0$. Due to (34),

$$J_1 \le \exp\left(-\lambda t\right) + \frac{tp_\sigma}{h}.$$

Substituting to (39),

$$I_1 \le \sum_{i=1}^{n}\left(\exp\left(-\lambda t\right) + \frac{tp_\sigma}{h}\right)\mathbb{E}\left[\exp\left(-\lambda h\sum_{k=1}^{\frac{K}{2}-1}\eta_{B,i,k}\right)\Big|\mathcal{G}_{B-1}\right].$$

Using the same arguments $\frac{K}{2} - 1$ times, we obtain

$$I_1 \le n\left(\exp\left(-\lambda t\right) + \frac{tp_\sigma}{h}\right)^{\frac{K}{2}} \tag{40}$$

for all $t \geq 0$. The analysis of $I_2^{K/2}$ a little bit more evolved. For all $1 \leq j \leq \frac{K}{2}$,

$$I_2^j := \sum_{i=1}^n \mathbb{E}\left[\exp\left(-\lambda\tau_s \sum_{k=1}^j \mu_{B,i,k}\right) \middle| \mathcal{G}_{B-1}\right]$$

$$= \sum_{i=1}^n \mathbb{E}\left[\underbrace{\mathbb{E}\left[\exp\left(-\lambda\tau_s \mu_{B,i,j}\right) | \mu_{B,i,j-1}, \ldots, \mu_{B,i,1}, \mathcal{G}_{B-1}\right]\exp\left(-\lambda\tau_s \sum_{k=1}^{j-1}\mu_{B,i,k}\right)}_{K_2:=} \middle| \mathcal{G}_{B-1}\right]$$

and $I_2^0 = n$. Let us define $u = \sum_{k=1}^{j-1}\mu_{B,i,k}$. If $u \geq \frac{d}{2}$, then

$$K_2 = \mathbb{E}\left[\exp\left(-\lambda\tau_s \mu_{B,i,j}\right) | \mu_{B,i,j-1}, \ldots, \mu_{B,i,1}, \mathcal{G}_{B-1}\right]\exp\left(-\lambda\tau_s u\right) \leq \exp\left(-\frac{\lambda\tau_s d}{2}\right)$$

Otherwise, if $u < \frac{d}{2}$, then, for all $t \geq 0$,

$$\mathbb{E}\left[\exp\left(-\lambda\tau_s \mu_{B,i,j}\right) | \mu_{B,i,j-1}, \ldots, \mu_{B,i,1}, \mathcal{G}_{B-1}\right]$$

$$\leq \exp\left(-\lambda t\right) + \mathbb{P}\left(\mu_{B,i,j} \leq \frac{t}{\tau_s} \middle| \mu_{B,i,j-1}, \ldots, \mu_{B,i,1}, \mathcal{G}_{B-1}\right)$$

$$\leq \exp\left(-\lambda t\right) + \frac{K\frac{t}{\tau_s}}{d - u - \frac{t}{\tau_s} + 1} \leq \exp\left(-\lambda t\right) + \frac{2Kt}{\tau_s d - 2t}$$

due to (35) and $u < \frac{d}{2}$. Combining both cases,

$$K_2 \leq \max\left\{\left(\exp\left(-\lambda t\right) + \frac{2Kt}{\tau_s d - 2t}\right)\exp\left(-\lambda\tau_s \sum_{k=1}^{j-1}\mu_{B,i,k}\right), \exp\left(-\frac{\lambda\tau_s d}{2}\right)\right\}$$

for all $t < \frac{\tau_s d}{2}$ and $u \geq 0$, and

$$I_2^j \leq \left(\exp\left(-\lambda t\right) + \frac{2Kt}{\tau_s d - 2t}\right)\sum_{i=1}^n \mathbb{E}\left[\exp\left(-\lambda\tau_s \sum_{k=1}^{j-1}\mu_{B,i,k}\right) \middle| \mathcal{G}_{B-1}\right] + n\exp\left(-\frac{\lambda\tau_s d}{2}\right)$$

$$= \left(\exp\left(-\lambda t\right) + \frac{2Kt}{\tau_s d - 2t}\right)I_2^{j-1} + n\exp\left(-\frac{\lambda\tau_s d}{2}\right), \tag{41}$$

where we use the inequality $\max\{a,b\} \leq a + b$ for all $a, b \geq 0$. Substituting (40) to (38),

$$\mathbb{E}\left[\exp\left(-\lambda \min_{i \in [n]}\beta_{B,i}\right) \middle| \mathcal{G}_{B-1}\right] \leq n\left(\exp\left(-\lambda t\right) + \frac{tp_\sigma}{h}\right)^{\frac{K}{2}} + I_2^{\frac{K}{2}},$$

where $t, \lambda \geq 0$ are free parameters. Taking $t = \frac{4 + \frac{2}{K}\log(2n)}{\lambda}$,

$$\mathbb{E}\left[\exp\left(-\lambda \min_{i \in [n]}\beta_{B,i}\right) \middle| \mathcal{G}_{B-1}\right] \leq n\left(\frac{e^{-4}}{(2n)^{2/K}} + \frac{(4 + \frac{2}{K}\log(2n))p_\sigma}{\lambda h}\right)^{\frac{K}{2}} + I_2^{\frac{K}{2}}.$$

Choosing $\lambda = e^4 (2n)^{2/K}(4 + \frac{2}{K}\log(2n))\max\left\{\frac{p_\sigma}{h}, \frac{p_K}{\tau_s}\right\}$,

$$\mathbb{E}\left[\exp\left(-\lambda \min_{i \in [n]}\beta_{B,i}\right) \middle| \mathcal{G}_{B-1}\right] \leq n\left(\frac{2e^{-4}}{(2n)^{2/K}}\right)^{\frac{K}{2}} + I_2^{\frac{K}{2}} = \frac{1}{2}\left(2e^{-4}\right)^{\frac{K}{2}} + I_2^{\frac{K}{2}}.$$

With this choice of $\lambda$ and $t$ in (41), we get

$$I_2^j \leq \left(\frac{3e^{-4}}{(2n)^{2/K}}\right)I_2^{j-1} + n\exp\left(-e^4 (2n)^{2/K}(4K + 2\log(2n))\right)$$

$$\leq \left(\frac{3e^{-4}}{(2n)^{2/K}}\right)I_2^{j-1} + e^{-4e^4 K} \leq n\left(\frac{3e^{-4}}{(2n)^{2/K}}\right)^j + 2e^{-4e^4 K}$$

for all $j \geq 1$ because $t \leq \frac{\tau_s d}{2e^4 (2n)^{2/K} K}$, $\lambda = \frac{4 + \frac{2}{K} \log(2n)}{t} \geq 2e^4 (2n)^{2/K} (4 + \frac{2}{K} \log(2n)) \frac{K}{\tau_s d}$. In the third inequality, we unrolled the recursion with $I_2^0 = n$ and use $\sum_{j=0}^{\infty} \left( \frac{3e^{-4}}{(2n)^{2/K}} \right)^j \leq 2$.

Finally, $I_2^{K/2} \leq \frac{1}{2} \left( 3e^{-4} \right)^{\frac{K}{2}} + 2e^{-2e^4 K} \leq \frac{1}{2} e^{-K}$ and

$$\mathbb{E} \left[ \exp \left( -\lambda \min_{i \in [n]} \beta_{B,i} \right) \bigg| \mathcal{G}_{B-1} \right] \leq \frac{1}{2} \left( 2e^{-4} \right)^{\frac{K}{2}} + I_2^{K/2} \leq \frac{1}{2} e^{-K} + \frac{1}{2} e^{-K} \leq e^{-K}.$$

Substituting the last inequality to (37) and repeating the steps $B - 1$ more times, we get

$$\mathbb{P} \left( \sum_{b=1}^{B} \min_{i \in [n]} \beta_{b,i} \leq \bar{t} \right) \leq \exp \left( \lambda \bar{t} - BK \right).$$

It is left to take $\bar{t} = \frac{BK + \log \delta}{\lambda}$. $\qquad \square$

# F    MAIN THEOREM WITH WORKER-TO-SERVER COMMUNICATION

In this section, we extend the result of Theorem 4.2 by taking into account the communication time $\tau_\mathrm{w}$. However, in this section, we ignore the communication times from the server to the workers in the analysis, which will be sufficient to obtain an almost tight lower bound if combined with Theorem 4.2.

Tyurin et al. (2024) consider a similar setup with $\tau_\mathrm{s} = 0$. However, their protocol does not allow the workers to modify the iterate computed by the server and operates in the primal space. For instance, the workers are not allowed to run local steps. Moreover, $\bar{P}_i^k$ are fixed in their version of Protocol 1. We improve upon this in the following theorem:

**Theorem F.1.** *Let $L, \Delta, \varepsilon, \sigma^2, n, d, \tau_\mathrm{w}, \tau_\mathrm{s}, h > 0$ be any numbers such that $\varepsilon < c_1 L\Delta$ and $d \geq \frac{\Delta L}{c_2 \varepsilon}$. Consider Protocol 1. For all $i \in [n]$ and $k \geq 0$, compressor $\bar{\mathcal{C}}_i^k$ selects and transmits $\bar{P}_i^k$ uniformly random coordinates without replacement, scaled by any constants, where $\bar{P}_i^k \in \{0, \ldots, d\}$ may vary across each compressor. For any algorithm $A \in \mathcal{A}_\mathrm{zr}$, there exists a function $f : \mathbb{R}^d \to \mathbb{R}$ such that $f$ is $L$-smooth, $f(0) - \inf_{x\in\mathbb{R}^d} f(x) \leq \Delta$, exists a stochastic gradient oracle that satisfies Assumption 1.3, and $\mathbb{E}\left[\inf_{y \in S_t} \|\nabla f(y)\|^2\right] > \varepsilon$ for all*

$$
t \leq c_3 \times \frac{L\Delta}{\varepsilon \log(n+1)} \cdot \min\left\{ \max\left\{ \frac{h\sigma^2}{n\varepsilon}, \frac{\tau_\mathrm{w} d}{n}, \sqrt{\frac{h\sigma^2 \tau_\mathrm{w} d}{n\varepsilon}}, h, \tau_\mathrm{w} \right\}, \max\left\{ \frac{h\sigma^2}{\varepsilon}, h \right\} \right\},
$$

*where $S_t$ is the set of all possible points that can be constructed by $A$ up to time $t$ based on $I$ and $\{I_i\}$. The quantities $c_1$, $c_2$, and $c_3$ are universal constants.*

*Proof.* The proof closely follows the analysis from (Tyurin et al., 2024; Tyurin & Richtárik, 2024) and the proof of Theorem 4.2, but with some important modifications. In this proof, it is sufficient to work with (6) and Lemmas 2.1 and 2.2.

Let us fix $\lambda > 0$ and define the function $f : \mathbb{R}^d \to \mathbb{R}$ such that

$$
f(x) := \frac{L\lambda^2}{\ell_1} F_T\left(\frac{x_{[T]}}{\lambda}\right),
$$

where the function $F_T$ is given in (6) and $x_{[T]} \in \mathbb{R}^T$ is the vector with the first $T$ coordinates of $x \in \mathbb{R}^d$. Notice that the last $d - T$ coordinates are artificial.

First, we have to show that $f$ is $L$-smooth and $f(0) - \inf_{x\in\mathbb{R}^d} f(x) \leq \Delta$. Using Lemma 2.2,

$$
\|\nabla f(x) - \nabla f(y)\| = \frac{L\lambda}{\ell_1}\left\|\nabla F_T\left(\frac{x_{[T]}}{\lambda}\right) - \nabla F_T\left(\frac{y_{[T]}}{\lambda}\right)\right\| \leq L\lambda\left\|\frac{x_{[T]}}{\lambda} - \frac{y_{[T]}}{\lambda}\right\|
$$
$$
= L\left\|x_{[T]} - y_{[T]}\right\| \leq L\left\|x - y\right\| \quad \forall x, y \in \mathbb{R}^d.
$$

Taking

$$
T = \left\lfloor \frac{\Delta \ell_1}{L\lambda^2 \Delta^0} \right\rfloor,
$$

$$
f(0) - \inf_{x\in\mathbb{R}^d} f(x) = \frac{L\lambda^2}{\ell_1}(F_T(0) - \inf_{x\in\mathbb{R}^T} F_T(x)) \leq \frac{L\lambda^2 \Delta^0 T}{\ell_1} \leq \Delta.
$$

due to Lemma 2.2.

Next, we construct a stochastic gradient mapping. For our lower bound, we define

$$
[\nabla f(x;\xi)]_j := \nabla_j f(x)\left(1 + \mathbb{1}\left[j > \mathrm{prog}(x)\right]\left(\frac{\xi}{p_\sigma} - 1\right)\right) \quad \forall x \in \mathbb{R}^d, \tag{42}
$$

and let $\mathcal{D}_\xi = Bernoulli(p_\sigma)$ for all $j \in [n]$, where $p_\sigma \in (0, 1]$. We denote $[x]_j$ as the $j^\mathrm{th}$ coordinate of a vector $x \in \mathbb{R}^d$. We choose

$$
p_\sigma := \min\left\{\frac{L^2\lambda^2\gamma_\infty^2}{\sigma^2\ell_1^2}, 1\right\}.
$$

Then this mapping is unbiased and $\sigma^2$-variance-bounded. Indeed,

$$\mathbb{E}\left[[\nabla f(x, \xi)]_i\right] = \nabla_i f(x)\left(1 + \mathbb{1}\left[i > \text{prog}(x)\right]\left(\frac{\mathbb{E}\left[\xi\right]}{p_\sigma} - 1\right)\right) = \nabla_i f(x)$$

for all $i \in [d]$, and

$$\mathbb{E}\left[\|\nabla f(x; \xi) - \nabla f(x)\|^2\right] \le \max_{j \in [d]} |\nabla_j f(x)|^2 \, \mathbb{E}\left[\left(\frac{\xi}{p_\sigma} - 1\right)^2\right]$$

because the difference is non-zero only in one coordinate. Thus

$$\mathbb{E}\left[\|\nabla f(x, \xi) - \nabla f(x)\|^2\right] \le \frac{\|\nabla f(x)\|_\infty^2 (1 - p_\sigma)}{p_\sigma} = \frac{L^2 \lambda^2 \left\|F_T\left(\frac{x_{[T]}}{\lambda}\right)\right\|_\infty^2 (1 - p_\sigma)}{\ell_1^2 p_\sigma}$$

$$\le \frac{L^2 \lambda^2 \gamma_\infty^2 (1 - p_\sigma)}{\ell_1^2 p_\sigma} \le \sigma^2,$$

where we use Lemma 2.2.

Taking

$$\lambda = \frac{\sqrt{2\varepsilon}\ell_1}{L},$$

we ensure that

$$\|\nabla f(x)\|^2 = \frac{L^2 \lambda^2}{\ell_1^2}\left\|\nabla F_T\left(\frac{x_{[T]}}{\lambda}\right)\right\|^2 > 2\varepsilon \mathbb{1}\left[\text{prog}(x_{[T]}) < T\right] \tag{43}$$

for all $x \in \mathbb{R}^d$, where we use Lemma 2.1. Thus

$$T = \left\lfloor \frac{\Delta L}{2\varepsilon \ell_1 \Delta^0} \right\rfloor \tag{44}$$

and

$$p_\sigma = \min\left\{\frac{2\varepsilon \gamma_\infty^2}{\sigma^2}, 1\right\}.$$

Using the same reasoning as in Tyurin et al. (2024) and our Theorem 4.2, we define two sets of random variables. Let $\eta_{1,i}$ be the first computed stochastic gradient when the oracle draws a "successful" Bernoulli trial in (42) at worker $i$. Then,

$$\mathbb{P}\left(\eta_{1,i} \le t\right) \le \sum_{i=1}^{\lfloor t \rfloor} (1 - p_\sigma)^{i-1} p_\sigma \le p_\sigma \lfloor t \rfloor$$

for $t \ge 0$, and

$$\mathbb{P}\left(\eta_{1,i} \le t\right) \le \min\{p_\sigma \lfloor t \rfloor, 1\}$$

For all $i \in [n]$, the server receives a stream of coordinates from worker $i$. Let $\mu_{1,i}$ be the number of received coordinates by the server from worker $i$ until the moment when the index of the last received coordinate is 1. Let us define

$$p_d := \frac{2}{d}.$$

Similarly to the proof of Theorem 4.2 with $K = 1$ (see (29)),

$$\mathbb{P}\left(\mu_{1,i} \le t | \eta_{1,i}\right) = \sum_{j=1}^{\lfloor t \rfloor} \mathbb{P}\left(\mu_{1,i,1} = j\right) \le \frac{\lfloor t \rfloor}{d - t + 1}.$$

for all $t \le d$. Thus,

$$\mathbb{P}\left(\mu_{1,i} \le t | \eta_{1,i}\right) \le \begin{cases} \frac{\lfloor t \rfloor}{d-t+1}, & t \le \frac{d}{2} \\ 1, & t > \frac{d}{2} \end{cases} \le \begin{cases} \lfloor t \rfloor p_d, & t \le \frac{d}{2} \\ 1, & t > \frac{d}{2} \end{cases} \le \min\{2 \lfloor t \rfloor p_d, 1\}$$

for all $t \geq 0$.

There are two ways in which worker $i$ can discover the first coordinate. Either the worker is "lucky" and draws a successful Bernoulli random variable locally, or it gets to discover the first coordinate through the server. Thus, worker $i$ requires at least

$$y_{1,i} := \min\left\{h\eta_{1,i}, \min_{j \in [n], j \neq i}\{h\eta_{1,j} + \tau_{\mathrm{w}}\mu_{1,j}\}\right\} = \min_{j \in [n]}\{h\eta_{1,j} + \mathbb{1}[i \neq j]\tau_{\mathrm{w}}\mu_{1,j}\}$$

seconds because $h\eta_{1,i}$ is the minimal time to discover the first coordinate locally, and $\min_{j \in [n], j \neq i}\{h\eta_{1,j} + \tau_{\mathrm{w}}\mu_{1,j}\}$ is the minimal time to discover the first coordinate from other workers via the server, which can transmit it to worker $i$.

*Remark* F.2. The previous derivations hold for all $\tau_{\mathrm{s}} > 0$. If we start taking the communication time $\tau_{\mathrm{s}}$ into account, then $y_{1,i}$ may only increase. For all $\tau_{\mathrm{s}} > 0$, worker $i$ still requires at least $y_{1,i}$ seconds for the same reason that $h\eta_{1,i}$ is the minimal time to discover the first coordinate locally, and $\min_{j \in [n], j \neq i}\{h\eta_{1,j} + \tau_{\mathrm{w}}\mu_{1,j}\}$ is the minimal time to discover the first coordinate from other workers. If we start taking into account the communication time from $\tau_{\mathrm{s}}$, the lower bound

$$\min\left\{h\eta_{1,i}, \min_{j \in [n], j \neq i}\{h\eta_{1,j} + \tau_{\mathrm{w}}\mu_{1,j}\}\right\}$$

still holds.

Using the same reasoning, worker $i$ requires at least

$$y_{k,i} := \min_{j \in [n]}\{h\eta_{k,j} + \mathbb{1}[i \neq j]\tau_{\mathrm{w}}\mu_{k,j} + y_{k-1,j}\}$$

seconds to discover the $k^{\mathrm{th}}$ coordinate for all $k \geq 2$, where

$$\mathbb{P}\left(\eta_{k,i} \leq t | \mathcal{G}_{k-1}\right) \leq \min\{\lfloor t \rfloor p_{\sigma}, 1\} \tag{45}$$

for all $k \geq 1$, $i \in [n]$, and $t \geq 0$, and

$$\mathbb{P}\left(\mu_{k,i} \leq t | \eta_{k,i}, \mathcal{G}_{k-1}\right) \leq \min\{2 \lfloor t \rfloor p_d, 1\} \tag{46}$$

for all $k \geq 1$, $i \in [n]$, and $t \geq 0$, where $\mathcal{G}_{k-1}$ is the sigma-algebra generated by $\{\eta_{k',i}\}_{i \in [n], k' < k}$ and $\{\mu_{k',i}\}_{i \in [n], k' < k}$. Thus, the first possible time when the workers and the server can discover the $T^{\mathrm{th}}$ coordinate is

$$y_T := \min_{i \in [n]} y_{T,i}.$$

For this random variable, we prove the lemma below (see Section F.1).

**Lemma F.3.** *Let* $\{\eta_{k,i}\}_{i,k \geq 0}$ *and* $\{\mu_{k,i}\}_{i,k \geq 0}$ *be random variables such that*

$$\mathbb{P}\left(\eta_{k,i} \leq t | \mathcal{G}_{k-1}\right) \leq \min\{\lfloor t \rfloor p_{\sigma}, 1\} \tag{47}$$

*for all* $k \geq 1$, $i \in [n]$, *and* $t \geq 0$, *and*

$$\mathbb{P}\left(\mu_{k,i} \leq t | \eta_{k,i}, \mathcal{G}_{k-1}\right) \leq \min\{2 \lfloor t \rfloor p_d, 1\}, \tag{48}$$

*for all* $k \geq 1$, $i \in [n]$, *and* $t \geq 0$, *where* $\mathcal{G}_{k-1}$ *is the sigma-algebra generated by* $\{\eta_{k',i}\}_{i \in [n], k' < k}$ *and* $\{\mu_{k',i}\}_{i \in [n], k' < k}$. *Then*

$$\mathbb{P}\left(y_T \leq \bar{t}\right) \leq \delta$$

*with*

$$\bar{t} := \frac{T - \log n + \log \delta}{32 \log(8n)} \cdot \min\left\{\max\left\{\frac{h}{p_{\sigma}n}, \frac{\tau_{\mathrm{w}}}{p_d n}, \frac{\sqrt{h\tau_{\mathrm{w}}}}{\sqrt{p_{\sigma}p_d}n}, h, \tau_{\mathrm{w}}\right\}, \frac{h}{p_{\sigma}}\right\}, \tag{49}$$

*where*

$$y_T := \min_{i \in [n]} y_{T,i},$$

$$y_{k,i} := \min_{j \in [n]}\{h\eta_{k,j} + \mathbb{1}[i \neq j]\tau_{\mathrm{w}}\mu_{k,j} + y_{k-1,j}\}$$

*for all* $k \geq 1$, $i \in [n]$ *and* $y_{0,i} = 0$ *for all* $i \in [n]$.

Thus, with probability at least $1 - \delta$, any zero-respecting algorithm requires at least $\bar{t}$ seconds to discover the last coordinate. Due to (43),

$$\inf_{y \in S_t} \|\nabla f(y)\|^2 > 2\varepsilon \inf_{y \in S_t} \mathbb{1}\left[\mathrm{prog}(y_{[T]}) < T\right],$$

where $S_t$ is the set of all possible candidate points to be an $\varepsilon$–stationary point up to time $t$, which can be computed by $A$. Taking $\delta = \frac{1}{2}$,

$$\mathbb{E}\left[\inf_{y \in S_t} \|\nabla f(y)\|^2\right] > 2\varepsilon \mathbb{E}\left[\inf_{y \in S_t} \mathbb{1}\left[\mathrm{prog}(y_{[T]}) < T\right]\right] \geq \varepsilon,$$

for $t = \frac{1}{2}\bar{t}$ because $\mathrm{prog}(y_{[T]}) < T$ for all $y \in S_t$ with probability at least $\frac{1}{2}$. It is left to substitute all quantities:

$$t = \frac{\left\lfloor \frac{\Delta L}{2\varepsilon \ell_1 \Delta^0} \right\rfloor - \log n + \log \frac{1}{2}}{64 \log(8n)} \cdot \min\left\{\max\left\{\frac{h}{n}\max\left\{\frac{\sigma^2}{2\varepsilon\gamma_\infty^2}, 1\right\}, \frac{\tau_\mathrm{w}d}{2n}, \frac{\sqrt{hd\tau_\mathrm{w}}}{\sqrt{2n}}\sqrt{\frac{\sigma^2}{2\varepsilon\gamma_\infty^2}}, h, \tau_\mathrm{w}\right\}, h\max\left\{\frac{\sigma^2}{2\varepsilon\gamma_\infty^2}, 1\right\}\right\}.$$

Since $\ell_1, \Delta^0, \gamma_\infty$ are universal constants, assuming $\varepsilon < c_1 L\Delta$ for some small universal $c_1 > 0$, we get

$$t \geq c_3 \times \frac{L\Delta}{\varepsilon \log(n+1)} \cdot \min\left\{\max\left\{\frac{h\sigma^2}{n\varepsilon}, \frac{\tau_\mathrm{w}d}{n}, \sqrt{\frac{h\sigma^2\tau_\mathrm{w}d}{n\varepsilon}}, h, \tau_\mathrm{w}\right\}, \max\left\{\frac{h\sigma^2}{\varepsilon}, h\right\}\right\}$$

for some small universal $c_3 > 0$. Notice that we can take any dimension $d$ such that

$$d \geq T = \Theta\left(\frac{L\Delta}{\varepsilon}\right).$$

$\square$

### F.1 MAIN CONCENTRATION LEMMA

**Lemma F.3.** *Let $\{\eta_{k,i}\}_{i,k\geq 0}$ and $\{\mu_{k,i}\}_{i,k\geq 0}$ be random variables such that*

$$\mathbb{P}\left(\eta_{k,i} \leq t|\mathcal{G}_{k-1}\right) \leq \min\{\lfloor t \rfloor p_\sigma, 1\} \tag{47}$$

*for all $k \geq 1$, $i \in [n]$, and $t \geq 0$, and*

$$\mathbb{P}\left(\mu_{k,i} \leq t|\eta_{k,i}, \mathcal{G}_{k-1}\right) \leq \min\{2\lfloor t \rfloor p_d, 1\}, \tag{48}$$

*for all $k \geq 1$, $i \in [n]$, and $t \geq 0$, where $\mathcal{G}_{k-1}$ is the sigma-algebra generated by $\{\eta_{k',i}\}_{i\in[n],k'<k}$ and $\{\mu_{k',i}\}_{i\in[n],k'<k}$. Then*

$$\mathbb{P}\left(y_T \leq \bar{t}\right) \leq \delta$$

*with*

$$\bar{t} := \frac{T - \log n + \log \delta}{32 \log(8n)} \cdot \min\left\{\max\left\{\frac{h}{p_\sigma n}, \frac{\tau_\mathrm{w}}{p_d n}, \frac{\sqrt{h\tau_\mathrm{w}}}{\sqrt{p_\sigma p_d n}}, h, \tau_\mathrm{w}\right\}, \frac{h}{p_\sigma}\right\}, \tag{49}$$

*where*

$$y_T := \min_{i \in [n]} y_{T,i},$$

$$y_{k,i} := \min_{j \in [n]}\left\{h\eta_{k,j} + \mathbb{1}[i \neq j]\tau_\mathrm{w}\mu_{k,j} + y_{k-1,j}\right\}$$

*for all $k \geq 1$, $i \in [n]$ and $y_{0,i} = 0$ for all $i \in [n]$.*

*Proof.* Using the Chernoff method for any $s > 0$ and $k \geq 1$, we get

$$\mathbb{P}\left(y_k \leq \bar{t}\right) = \mathbb{P}\left(-sy_k \geq -s\bar{t}\right) = \mathbb{P}\left(e^{-sy_k} \geq e^{-s\bar{t}}\right) \leq e^{s\bar{t}}\mathbb{E}\left[e^{-sy_k}\right]$$

$$= e^{s\bar{t}}\mathbb{E}\left[\exp\left(-s\min_{j\in[n]}y_{k,j}\right)\right] = e^{s\bar{t}}\mathbb{E}\left[\max_{j\in[n]}\exp\left(-sy_{k,j}\right)\right].$$

Bounding the maximum by the sum,

$$\mathbb{P}\left(y_k \le t\right) \le e^{st}\sum_{j=1}^{n}\mathbb{E}\left[\exp\left(-sy_{k,j}\right)\right] \le ne^{st}\max_{j\in[n]}\mathbb{E}\left[\exp\left(-sy_{k,j}\right)\right]. \tag{50}$$

We focus on the last exponent separately. For all $i \in [n]$,

$$\begin{aligned}
\mathbb{E}\left[\exp\left(-sy_{k,i}\right)\right] &= \mathbb{E}\left[\exp\left(-s\min_{j\in[n]}\{h\eta_{k,j} + \mathbb{1}[i \ne j]\tau_{\mathrm{w}}\mu_{k,j} + y_{k-1,j}\}\right)\right] \\
&= \mathbb{E}\left[\max_{j\in[n]}\exp\left(-s\left(h\eta_{k,j} + \mathbb{1}[i \ne j]\tau_{\mathrm{w}}\mu_{k,j} + y_{k-1,j}\right)\right)\right] \\
&\le \sum_{j=1}^{n}\mathbb{E}\left[\exp\left(-s\left(h\eta_{k,j} + \mathbb{1}[i \ne j]\tau_{\mathrm{w}}\mu_{k,j} + y_{k-1,j}\right)\right)\right] \\
&= \sum_{j=1}^{n}\mathbb{E}\left[\underbrace{\mathbb{E}\left[\exp\left(-s\left(h\eta_{k,j} + \mathbb{1}[i \ne j]\tau_{\mathrm{w}}\mu_{k,j}\right)\right)|\,\mathcal{G}_{k-1}\right]}_{I_1 :=}\exp\left(-sy_{k-1,j}\right)\right]
\end{aligned} \tag{51}$$

Considering the inner expectation separately:

$$\begin{aligned}
I_1 &= \mathbb{E}\left[\exp\left(-s\left(h\eta_{k,j} + \mathbb{1}[i \ne j]\tau_{\mathrm{w}}\mu_{k,j}\right)\right)|\,\mathcal{G}_{k-1}\right] \\
&\le \exp\left(-st\right) + \mathbb{P}\left(h\eta_{k,j} + \mathbb{1}[i \ne j]\tau_{\mathrm{w}}\mu_{k,j} \le t|\mathcal{G}_{k-1}\right)
\end{aligned}$$

for all $t \ge 0$. Using the properties of condition expectations,

$$\begin{aligned}
I_1 &\le \exp\left(-st\right) + \mathbb{P}\left(h\eta_{k,j} \le t, \mathbb{1}[i \ne j]\tau_{\mathrm{w}}\mu_{k,j} \le t|\mathcal{G}_{k-1}\right) \\
&= \exp\left(-st\right) + \mathbb{E}\left[\mathbb{1}\left[h\eta_{k,j} \le t\right]\mathbb{1}\left[\mathbb{1}[i \ne j]\tau_{\mathrm{w}}\mu_{k,j} \le t\right]|\,\mathcal{G}_{k-1}\right] \\
&= \exp\left(-st\right) + \mathbb{E}\left[\mathbb{E}\left[\mathbb{1}\left[\mathbb{1}[i \ne j]\tau_{\mathrm{w}}\mu_{k,j} \le t\right]|\,\eta_{k,j},\mathcal{G}_{k-1}\right]\mathbb{1}\left[h\eta_{k,j} \le t\right]|\,\mathcal{G}_{k-1}\right] \\
&= \exp\left(-st\right) + \mathbb{E}\left[\mathbb{P}\left(\mathbb{1}[i \ne j]\tau_{\mathrm{w}}\mu_{k,j} \le t|\eta_{k,j},\mathcal{G}_{k-1}\right)\mathbb{1}\left[h\eta_{k,j} \le t\right]|\,\mathcal{G}_{k-1}\right].
\end{aligned}$$

If $i = j$, then we bound the probability by 1 and get

$$\begin{aligned}
I_1 &\le \exp\left(-st\right) + \mathbb{E}\left[\mathbb{1}\left[h\eta_{k,j} \le t\right]|\,\mathcal{G}_{k-1}\right] \\
&= \exp\left(-st\right) + \mathbb{P}\left(h\eta_{k,j} \le t|\mathcal{G}_{k-1}\right) \\
&\le \exp\left(-st\right) + \left\lfloor\frac{t}{h}\right\rfloor p_\sigma,
\end{aligned}$$

for all $t \ge 0$, where we use (47). Otherwise, if $i \ne j$, using (48) and (47),

$$\begin{aligned}
I_1 &\le \exp\left(-st\right) + \mathbb{E}\left[\min\left\{1, 2\left\lfloor\frac{t}{\tau_{\mathrm{w}}}\right\rfloor p_d\right\}\mathbb{1}\left[h\eta_{k,j} \le t\right]\Big|\,\mathcal{G}_{k-1}\right] \\
&= \exp\left(-st\right) + \min\left\{1, 2\left\lfloor\frac{t}{\tau_{\mathrm{w}}}\right\rfloor p_d\right\}\mathbb{P}\left(h\eta_{k,j} \le t|\mathcal{G}_{k-1}\right) \\
&\le \exp\left(-st\right) + \min\left\{1, 2\left\lfloor\frac{t}{\tau_{\mathrm{w}}}\right\rfloor p_d\right\}\min\left\{1, \left\lfloor\frac{t}{h}\right\rfloor p_\sigma\right\}
\end{aligned}$$

for all $t \ge 0$. Substituting the inequalities to (51),

$$\begin{aligned}
\mathbb{E}\left[\exp\left(-sy_{k,i}\right)\right] = \sum_{j\ne i}&\left(\exp\left(-st\right) + \min\left\{1, 2\left\lfloor\frac{t}{\tau_{\mathrm{w}}}\right\rfloor p_d\right\}\min\left\{1, \left\lfloor\frac{t}{h}\right\rfloor p_\sigma\right\}\right)\mathbb{E}\left[\exp\left(-sy_{k-1,j}\right)\right] \\
&+ \left(\exp\left(-st\right) + \min\left\{1, \left\lfloor\frac{t}{h}\right\rfloor p_\sigma\right\}\right)\mathbb{E}\left[\exp\left(-sy_{k-1,i}\right)\right].
\end{aligned}$$

for all $i \in [n]$ and $t \ge 0$. Thus,

$$\max_{i\in[n]}\mathbb{E}\left[\exp\left(-sy_{k,i}\right)\right]$$

$$\leq \left[ (n-1) \left( \exp\left(-st\right) + \min\left\{1, 2\left\lfloor\frac{t}{\tau_{\mathrm{w}}}\right\rfloor p_d\right\} \min\left\{1, \left\lfloor\frac{t}{h}\right\rfloor p_\sigma\right\}\right) + \left(\exp\left(-st\right) + \min\left\{1, \left\lfloor\frac{t}{h}\right\rfloor p_\sigma\right\}\right)\right]$$

$$\times \max_{i\in[n]} \mathbb{E}\left[\exp\left(-sy_{k-1,i}\right)\right]$$

$$= \left[ n\exp\left(-st\right) + (n-1)\left(\min\left\{1, 2\left\lfloor\frac{t}{\tau_{\mathrm{w}}}\right\rfloor p_d\right\}\min\left\{1, \left\lfloor\frac{t}{h}\right\rfloor p_\sigma\right\}\right) + \min\left\{1, \left\lfloor\frac{t}{h}\right\rfloor p_\sigma\right\}\right] \max_{i\in[n]} \mathbb{E}\left[\exp\left(-sy_{k-1,i}\right)\right].$$

Taking $s = \frac{\log(8n)}{t}$, we get

$$\max_{i\in[n]} \mathbb{E}\left[\exp\left(-sy_{k,i}\right)\right]$$

$$\leq \left[ \frac{1}{8} + \underbrace{(n-1)\left(\min\left\{1, 2\left\lfloor\frac{t}{\tau_{\mathrm{w}}}\right\rfloor p_d\right\}\min\left\{1, \left\lfloor\frac{t}{h}\right\rfloor p_\sigma\right\}\right)}_{I_2:=} + \underbrace{\min\left\{1, \left\lfloor\frac{t}{h}\right\rfloor p_\sigma\right\}}_{I_3:=}\right] \max_{i\in[n]} \mathbb{E}\left[\exp\left(-sy_{k-1,i}\right)\right].$$

$$(52)$$

Next, we take $t = \min\{t_1, t_2\}$, where

$$t_1 := \max\left\{\frac{h}{32p_\sigma n}, \frac{\tau_{\mathrm{w}}}{32p_d n}, \frac{\sqrt{h\tau_{\mathrm{w}}}}{32\sqrt{p_\sigma p_d}n}, \frac{h}{32}, \frac{\tau_{\mathrm{w}}}{32}\right\},$$

and

$$t_2 := \frac{h}{32p_\sigma}$$

to ensure that

$$I_3 \leq \min\left\{1, \frac{t_2 p_\sigma}{h}\right\} \leq \frac{1}{16}.$$

There are five possible values of $t_1$.

If $t_1 = \frac{h}{32p_\sigma n}$, then

$$I_2 \leq (n-1)\min\left\{1, \frac{t_1 p_\sigma}{h}\right\} \leq \frac{1}{16},$$

If $t_1 = \frac{\tau_{\mathrm{w}}}{32p_d n}$, then

$$I_2 \leq (n-1)\min\left\{1, \frac{2t_1 p_d}{\tau_{\mathrm{w}}}\right\} \leq \frac{1}{16}.$$

If $t_1 = \frac{\sqrt{h\tau_{\mathrm{w}}}}{32\sqrt{p_\sigma p_d}n}$, then

$$I_2 \leq (n-1)\frac{2t_1^2 p_d p_\sigma}{\tau_{\mathrm{w}} h} \leq \frac{1}{16}.$$

If $t_1 = \frac{h}{32}$, then

$$I_2 \leq (n-1)\min\left\{1, \left\lfloor\frac{t_1}{h}\right\rfloor p_\sigma\right\} = 0.$$

Finally, if $t_1 = \frac{\tau_{\mathrm{w}}}{32}$, then

$$I_2 \leq (n-1)\min\left\{1, 2\left\lfloor\frac{t_1}{\tau_{\mathrm{w}}}\right\rfloor p_d\right\} = 0.$$

Thus, using (52), we obtain

$$\max_{i\in[n]} \mathbb{E}\left[\exp\left(-sy_{k,i}\right)\right] \leq \left[\frac{1}{8} + \frac{1}{16} + \frac{1}{16}\right] \max_{i\in[n]} \mathbb{E}\left[\exp\left(-sy_{k-1,i}\right)\right] \leq e^{-1} \max_{i\in[n]} \mathbb{E}\left[\exp\left(-sy_{k-1,i}\right)\right]$$

for our choice of $t$. Unrolling the recursion and using $y_{0,i} = 0$ for all $i\in[n]$,

$$\max_{i\in[n]} \mathbb{E}\left[\exp\left(-sy_{k,i}\right)\right] \leq e^{-k}.$$

We substitute it to (50), to get

$$\mathbb{P}\left(y_k \leq \bar{t}\right) \leq e^{s\bar{t}+\log n - k}.$$

It is left to choose $\bar{t} = \frac{k - \log n + \log\delta}{s}$. $\qquad\square$

