# OpenReview forum: "Proving the Limited Scalability of Centralized Distributed Optimization via a New Lower Bound Construction"
_ICLR.cc/2026/Conference — ICLR 2026 Poster_

### Official Review · Reviewer_Neht · 2025-10-24

**Soundness:** 3
**Presentation:** 3
**Contribution:** 3
**Rating:** 6
**Confidence:** 3

**Summary:**

This paper investigates the fundamental limitations of centralized distributed optimization within the federated learning framework. It establishes a new lower bound on the time complexity for finding an ε-stationary point of smooth non-convex functions. A key aspect of the work is the explicit inclusion of bidirectional communication costs, particularly the server-to-worker  communication time. The paper reveals fundamental limitations in scaling distributed optimization.

**Strengths:**

1.The explicit consideration of bidirectional communication costs (τs and τw) is a powerful and novel contribution, reflecting a more realistic model of distributed systems.
2.The paper makes a significant theoretical contribution by constructing a new lower bound that clarifies the scalability limits of a specific class of optimization algorithms.
3.The construction of a novel "worst-case" function and the associated proof framework is a strong technical achievement.

**Weaknesses:**

1.The paper's discussion on the practical implications of its pessimistic results for unbiased compressors is too brief. It would be significantly strengthened by a more detailed exploration of how this points to the potential superiority of biased compressors in real-world scenarios.
2.The practical takeaways for system designers and practitioners are not stated explicitly enough. The paper should offer clearer guidance on how these theoretical results should inform practical decisions (e.g., algorithm selection based on network characteristics).
3.The scope is limited to unbiased compressors, while many state-of-the-art methods use biased ones. The paper should better position itself regarding this gap between its theoretical setting and common practice.
4.The notation is dense. A table of key symbols and their definitions in the appendix would improve the paper's readability.

**Questions:**

1.Given the pessimistic results for unbiased compressors when τs > 0, does this imply that biased compressors (e.g., TopK) are a fundamentally better choice in practical scenarios with communication bottlenecks?
2.Can the current analysis framework be extended to provide lower bounds for biased compressors? What are the main challenges in doing so, or would a completely new approach be required?
3.How should a practitioner interpret these results when designing a distributed system? For instance, in a network with symmetric communication costs (τs ≈ τw), does your lower bound suggest that investing in the studied forms of compression is futile and methods like Batch Synchronized SGD are preferable?

---

> ### Author Response · Authors · 2025-11-13
>
> Thank you for the review! Let us respond to the comments:
>
> > The practical takeaways for system designers and practitioners are not stated explicitly enough. The paper should offer clearer guidance on how these theoretical results should inform practical decisions (e.g., algorithm selection based on network characteristics).
>
> > How should a practitioner interpret these results when designing a distributed system? For instance, in a network with symmetric communication costs (τs ≈ τw), does your lower bound suggest that investing in the studied forms of compression is futile and methods like Batch Synchronized SGD are preferable?
>
> The takeaways are in the contribution section: when $\tau_s \sim \tau_w,$ which is the common case, one should use non-compressed methods such as Batch QSGD or a non-distributed SGD method on one of the workers. In other words, methods with compression techniques offer no theoretical advantage (in the worst case).
>
> On the other hand, if $\tau_s \leq \tau_w,$ then compression techniques might help. In this case, the compression techniques can help on the workers side in the regimes when $\frac{\tau_{\textnormal{w}} d}{n} + \sqrt{\frac{d \tau_{\textnormal{w}} h \sigma^2}{n \varepsilon}}$ is larger than ${\tau_{\textnormal{s}} d}$, due to the former scaling with $n$.
> Please consider the clarifications that we have added to Section 1.2 of the PDF.
>
> > The notation is dense. A table of key symbols and their definitions in the appendix would improve the paper's readability.
>
> The notations section is in Section C.1. We've expanded it with new common notations.
>
> > Given the pessimistic results for unbiased compressors when τs > 0, does this imply that biased compressors (e.g., TopK) are a fundamentally better choice in practical scenarios with communication bottlenecks?
>
> Notice that the family of unbiased compressors contains the family of biased compressors [1]. Therefore, our lower bounds also apply to methods that use biased compressors, in the sense that there exists a ``worst-case'' compressor for which these methods cannot achieve a convergence rate faster than the lower bound in Theorem 1.6. We've clarified it in the conclusion section of the PDF.
>
> > Can the current analysis framework be extended to provide lower bounds for biased compressors? What are the main challenges in doing so, or would a completely new approach be required?
>
> Our construction works also for methods that work with biased compressors for the same reason as before: biased compressors include unbiased compressors. The scaled version of RandK (random sparsifier) is also a biased compressor if one scales it by $1 / (\omega + 1),$ where $\omega$ is defined in Definition 1.5. We conjecture that the lower bounds also hold for the entire family of unbiased and biased compressors, but proving this would require more sophisticated constructions. Extending the result to all compressors is an open and challenging question. However, even for random sparsifiers, the result is non-obvious and requires new constructions.
>
> ---
>
> **We hope that we have answered all the weaknesses and questions. If you have any questions, please let us know.**
>
> [1]: Aleksandr Beznosikov, Samuel Horvath, Peter Richtarik, and Mher Safaryan. On biased compression for distributed learning. arXiv preprint arXiv:2002.12410, 2020.

---

### Official Review · Reviewer_vzMo · 2025-11-02

**Soundness:** 4
**Presentation:** 3
**Contribution:** 4
**Rating:** 6
**Confidence:** 4

**Summary:**

This paper presents new lower bounds for centralized distributed non-convex optimization of smooth functions. This is an active area, where convergence guarantees for finding approximate stationary points using first-order stochastic oracles have been studied extensively in the past several years. The key difference in this paper's setting is that, instead of examining oracle, communication, or iteration complexity, the paper focuses on time complexity in a model where each machine has the same computation speed and potentially different up (to the server) and down (from the server) communication times. The paper considers the homogeneous setting, for which it is more challenging to establish lower bounds, as one can not benefit from the usual round complexity constructions, which force machines to depend on each other for growing the coordinate span of their model.

The most remarkable, and perhaps astonishing, result that the paper shows is that when the up and down communication times are comparable, the optimal time complexity is attained by the best of (uncompressed) mini-batch SGD and single-machine SGD. Given the attention this research area has received, a plethora of algorithms have been devised and analyzed, including compression algorithms. However, the fact that one of the two simplest algorithms is min-max optimal provides a firm foundation for the optimization theory in this area. Notably, this result is comparable to the seminal work of [Woodworth et al.](https://arxiv.org/abs/2102.01583), which also reveals the same dichotomy for homogeneous smooth convex optimization, but without considering computation and communication times.

Additionally, the paper presents a lower bound in the case where up and down communication require different times. In that setting, the bound is still matched by either one of the most natural compression algorithms, mini-batch QSGD or single machine SGD. The hard instance for the lower bound follows the serial hard instance due to Carmon et al., which in turn follows the chain function of Nesterov. The authors make the original hard instance more complex by altering the process of discovering the next coordinate.

Overall, besides minor writing issues, I don't have any significant concerns with the paper and recommend accepting it. Once the authors address the writing issues, I am happy to increase my score.

**Strengths:**

The paper is well-written; it provides a comprehensive survey of the current landscape of results and algorithms, ultimately highlighting the gap in existing rates when it comes to time complexity, thereby motivating the central question of the paper. The paper also does a good job of summarizing the technical preliminaries and clearly expresses the novel technical idea. As mentioned above, the dichotomy between mini-batch and single machine SGD is a somewhat surprising result in the non-convex setting. It provides clarity on existing gaps and guides future research.

**Weaknesses:**

1. I believe the paper can do a more thorough job of reviewing the relevant literature. For instance, the morally most similar paper due to [Woodworth et al.](https://arxiv.org/abs/2102.01583) is not discussed. Similarly, several other interesting related threads are not discussed, such as multi-point oracles and variance reduction, which present natural improvements in the non-convex setting and where, in the oracle complexity model, (almost) min-max rates [are known](https://openreview.net/forum?id=SNElc7QmMDe).
2. While the lower bound of the paper is surprising, I believe it is essential to highlight that the lower bound itself does not indicate or explain the "limited scalability of centralized distributed optimization". Firstly, the lower bound considers a specific setting of homogeneous non-convex optimization. It is very well possible that under additional assumptions, such as higher-order smoothness and/or some notion of quasi-convexity, we might expect different results and optimal algorithms. I would encourage the authors to read the future directions presented by [Woodworth et al.](https://arxiv.org/abs/2102.01583), which indeed motivated many exciting follow-up results. I recommend that the authors temper the conclusions to be drawn from the results, and instead use these to either motivate why other settings might be interesting to study or show empirical evidence that centralized distributed optimization has indeed plateaued (the latter is very unlikely to be the case).
3. It is essential to underline that while in optimization theory the variance reduction due to averaging across the clients is usually the only justification for collaboration, that is a very narrow view. In a sense, the model under which such rates are provided is actually too restrictive to demonstrate the other benefits of collaboration. For instance, in the same vein as the hard instance of this paper, collaboration can help clients discover parts of the loss landscape that are inaccessible to individual clients. This effect does not appear in the stochastic optimization setup where fresh samples are used, and there is no distribution shift between the training and test distributions. Both of these effects are unavoidable in practice. Similarly, there is ample evidence that collaboration can have a regularization effect. This has been studied in the context of local update algorithms ([1](https://openreview.net/forum?id=dOoPSZFDiRB), [Section 4.3](https://arxiv.org/pdf/2507.00195)), but it may also be true in general for other forms of collaboration.

Overall, in the last two points, I aim to emphasize that it is crucial to note that the paper examines a specific theoretical model, which does provide some evidence for what we might observe in real experiments. However, ultimately, that is a particular model, which can not capture all aspects of empirical training.

4. I believe the presentation and definition of distributed zero respecting algorithms (whole of section 2.1) can be made much more crisp by introducing an oracle model. I recommend that the authors check the definitions in [Patel et al.](https://openreview.net/forum?id=SNElc7QmMDe) and other similar papers. Protocol 1 is quite confusing, and I especially don't like how the authors present two parallel for loops sequentially. I believe it is ambiguous to say what it means to calculate the next point, based on some information. This can be made more rigorous by considering a coordinate span, like the papers in this area do.

**Questions:**

1. Because section 2.1 is not super clear, does the class of algorithms considered here allow simultaneous queries (multiple queries at the same random seed) or not? What about local update algorithms? Can the queries be adaptive to the information on each machine locally or not? I would like the authors to comment on the results of [Patel et al.](https://openreview.net/pdf?id=SNElc7QmMDe), which may provide better upper bounds when simultaneous queries are permitted.
2. In the vein of the previous question, can the authors use their hard instance to give better lower bounds in the setting when computation and communication times are not critical, and we are looking at oracle complexity? Specifically, I want to understand whether the new technique can actually demonstrate that the convergence rate of SCAFFOLD and MB-SGD is tight in the oracle complexity sense. For more context, see [Table 1](https://openreview.net/pdf?id=SNElc7QmMDe) of Patel et al. and note that there is no lower bound for the single query model (their Theorem 3.2 can not demarcate between single and multiple queries).
3. How is the result in this paper morally different from [Woodworth et al.](https://arxiv.org/abs/2102.01583)? Should I view the consequence of the result as just the time-complexity version of their result? If so, can the authors comment on whether they believe non-convexity is actually critical to their construction? If not, is it possible to get a similar dichotomy between mini-batch and single-machine SGD in the convex setting as well? That would complete the picture in the homogeneous setting.

---

> ### Author Response · Authors · 2025-11-13
> **Official Comment by Authors (Part 1)**
>
> Thank you for your positive review! We now respond to the weaknesses:
>
> > I believe the paper can do a more thorough job of reviewing the relevant literature. ...
>
> Agree. See Section B in the updated PDF, where we have added a broader discussion of related work.
>
> > While the lower bound of the paper is surprising, I believe it is essential to highlight that the lower bound itself does not indicate or explain the "limited scalability of centralized distributed optimization". ...
>
> Also agree. In the conclusion, we have clarified that the lower bound can be potentially broken under additional assumptions.
>
> > It is essential to underline that while in optimization theory the variance reduction due to averaging across the clients is usually the only justification for collaboration, that is a very narrow view. ...
>
> This may all be true. However, our goal is not to solve or provide lower bounds for every possible problem, including the finite-sum setting or generalization capabilities. For instance, analyzing the regularization effect of distributed optimization is important but an orthogonal task. We focus on the time performance of algorithms, which is a fundamental question in the design of optimization methods. We agree that there are many unexplored and unsolved problems in distributed optimization.
>
> > I believe the presentation and definition of distributed zero respecting algorithms (whole of section 2.1) can be made much more crisp by introducing an oracle model.
>
> We've incorporated and improved the presentation of Protocol 1 and made it more formal in the updated PDF.
>
> ---
>
> **Please consider the updated PDF where we have incorporated your suggestions.**

---

> ### Author Response · Authors · 2025-11-13
> **Official Comment by Authors (Part 2)**
>
> We now respond to the questions:
>
> > Because section 2.1 is not super clear, does the class of algorithms considered here allow simultaneous queries (multiple queries at the same random seed) or not?
>
> In Protocol 1, we do not consider multiple queries. However, similarly to Theorem 1 by [1], additional queries will not help algorithms to break the lower bound. Please consider our Remark E.1 where we explain that multiple queries do not help. Notice that multiple queries help only under an additional assumption called the mean-squared property (L-mean smoothness; see [2]) and using variance reduction techniques.
>
> > What about local update algorithms? Can the queries be adaptive to the information on each machine locally or not?
>
> Yes, the local updates are allowed, as an algorithm can be adaptive to the local information. The only restriction is that the support of the query points belongs to the support of the local information (see the updated Protocol 1).
>
> > I would like the authors to comment on the results of Patel et al., which may provide better upper bounds when simultaneous queries are permitted.
>
> The reason why [2] get better bounds is that they use variance reduction, which requires an additional assumption called the mean-squared property (L-mean smoothness in their paper). We don’t assume that assumption, so there is no contradiction. To design a lower bound for the Patel et al.’s setting, one may try to modify our construction in the same way it was done in Section 3.3 of [1].
>
> > In the vein of the previous question, can the authors use their hard instance to give better lower bounds in the setting when computation and communication times are not critical, and we are looking at oracle complexity? Specifically, I want to understand whether the new technique can actually demonstrate that the convergence rate of SCAFFOLD and MB-SGD is tight in the oracle complexity sense. ...
>
> In terms of time complexities, it was done in [4,5]; consider their results with $h_i = h$ and $\tau_{i \to j} = \tau.$ Note that the new hard instance is required for the problems and methods that utilize compression. Without compressions, it might be possible to get tight time complexity lower bounds with the hard instance by [6].
>
> > How is the result in this paper morally different from Woodworth et al.? Should I view the consequence of the result as just the time-complexity version of their result?
>
> Our paper and paper [3] are related to, but not directly comparable with ours, since we analyze the limited scalability of improving both stochastic noise and communication complexity through *compression techniques*. The latter wasn't considered in [3].
>
> > If so, can the authors comment on whether they believe non-convexity is actually critical to their construction? If not, is it possible to get a similar dichotomy between mini-batch and single-machine SGD in the convex setting as well? That would complete the picture in the homogeneous setting.
>
> This is a good open question. Getting the same result in the convex setting is an important future direction.
>
> ---
>
> **Thank you once again for your useful comments and questions. We have used them to improve our paper; please consider the updated PDF. If you believe that we have addressed all the comments, we kindly ask you to reconsider the score. If you have any questions, please let us know.**
>
> ---
>
> [1]: Yossi Arjevani, Yair Carmon, John C Duchi, Dylan J Foster, Nathan Srebro, and Blake Woodworth. Lower bounds for non-convex stochastic optimization. Mathematical Programming, pp. 1–50, 2022.
>
> [2]: Kumar Kshitij Patel, Lingxiao Wang, Blake Woodworth, Brian Bullins, and Nati Srebro. Towards optimal communication complexity in distributed non-convex optimization. NeurIPS 2022
>
> [3]: Blake E Woodworth, Brian Bullins, Ohad Shamir, and Nathan Srebro. The min-max complexity of distributed stochastic convex optimization with intermittent communication. COLT 2021.
>
> [4]: Alexander Tyurin and Peter Richtarik. Optimal time complexities of parallel stochastic optimization methods under a fixed computation model. NeurIPS 2023.
>
> [5]: Alexander Tyurin and Peter Richtarik. On the optimal time complexities in decentralized stochastic
> asynchronous optimization. NeurIPS 2024
>
> [6]: Yair Carmon, John C Duchi, Oliver Hinder, and Aaron Sidford. Lower bounds for finding stationary points i. Mathematical Programming, 184(1):71–120, 2020.

---

### Official Review · Reviewer_9bv8 · 2025-11-09

**Soundness:** 3
**Presentation:** 3
**Contribution:** 3
**Rating:** 6
**Confidence:** 3

**Summary:**

This paper presents lower bounds for centralized federated learning under $L$-smoothness, where communication time is important. It contains a new construction of lower bounds with theoretical guarantees.

 I think this paper may be accepted.

**Strengths:**

1) This paper is well written, and all the results are clarified.

2) This paper provides a new construction for lower bounds.

3) The result of this paper provides theoretical evidence and a quality comparison with previous works.

**Weaknesses:**

**Typos:**

1). p. 5 line 228. It seems “quartic” $\to$ “quadratic”

2). P. 7 line 377. I think, $F_T \to F_{T, K, a}$

3). P. 22 line 1146. $j \to k$

**Questions:**

I checked all the theorems, except for key Lemma D.1. Therefore, I don't have any major or minor comments, and for this reason, I marked a low level of confidence.

---

> ### Author Response · Authors · 2025-11-13
>
> Thank you for your positive review!
>
> We’ve already fixed the typos in the updated PDF.

---

### Official Review · Reviewer_2Z8R · 2025-11-10

**Soundness:** 3
**Presentation:** 2
**Contribution:** 3
**Rating:** 4
**Confidence:** 2

**Summary:**

In the paper the authors discuss the influence of both worker to server (w2s) and server to worker (s2w) communications on the overall complexity of solving the distributed optimization problem. The authors propose a new worst-case function, that is required to prove the lower bounds for this setup.

**Strengths:**

1)Addressing not only w2s, but also s2w communications is important and not usually done in optimization.

2)Modifying the worst-case function and using novel techniques to prove the lower bounds.

**Weaknesses:**

1)The class of zero-respecting protocols is quite restricting, as it does not contain sketchings, dithered quantizations, and so on.

2)Theorems 4.2 and E.1 seems to be proven only for RandK compressors, rather than for any unbiased zero-respecting.

**Questions:**

1)Theorem 4.2 contains only $\tau_s$, but no $\tau_w$. Theorem E.1, on the contrary, contains $\tau_w$, but no $\tau_s$. Could the authors clarify, how they correctly combine lower bound, that involve both s2w and w2s communications?

2)Could authors specify more, how does the heterogeneous setup fail under the consideration of s2w communications?

**Details Of Ethics Concerns:**

No additional ethical concerns.

---

> ### Author Response · Authors · 2025-11-13
>
> Thank you. We now respond to the weaknesses and questions.
>
> > The class of zero-respecting protocols is quite restricting, as it does not contain sketchings, dithered quantizations, and so on.
>
> This is a standard class of methods considered in the community [1,2,3] and many other papers. This family includes SGD-like, Adam-like, and asynchronous SGD-like methods.
>
> > Theorems 4.2 and E.1 seems to be proven only for RandK compressors, rather than for any unbiased zero-respecting.
>
> This is true. However, this is standard practice for taking the “worst-case” compressors from the family (similarly to taking the “worst-case” functions (Carmon et al., 2020; Nesterov, 2018)). Moreover, due to the uncertainty principle (Safaryan et al., 2022), all unbiased compressors exhibit variance and communication cost comparable to those of the RandK sparsifier. In the worst case, we expect the lower bound to hold for all compressors.
>
> > Theorem 4.2 contains only $\tau_s$, but no $\tau_w$. Theorem E.1, on the contrary, contains $\tau_w$, but no $\tau_s$. Could the authors clarify, how they correctly combine lower bound, that involve both s2w and w2s communications?
>
> Combining the lower bound from Theorems 4.2 and F.1, the lower bound is the maximum of the results:
> $$(1) = \max[ \min[A,B], \min[C,B]],$$
> where $A, B, C$ are the corresponding formulas that depend on $h, \sigma^2,$ and so on, the theorems share the same $B = \Theta(h \frac{L \Delta}{\varepsilon} + h \frac{L \Delta \sigma^2}{\varepsilon^2}).$ Thus, if $B < A$ or $B < C,$ then
> $$(1) \geq \Omega(B).$$
> Otherwise, if $B \geq A$ and $B \geq C,$ then
> $$(1) \geq \max[A, C] \geq \Omega(A + C).$$ Thus, we can conclude that
> $$(1) \geq \Omega(\min[A + C, B])$$
> and get the result from Theorem 1.6.
>
> > Could authors specify more, how does the heterogeneous setup fail under the consideration of s2w communications?
>
> The heterogeneous setting is more general and also obeys our new lower bounds (consider the case $f_i = f$). Thus, our main result, Theorem 1.6, also holds in the case of heterogeneous functions, and the dependence on $\tau_{s}$ in the lower bound remains the same. Therefore, the heterogeneous setup does not fail the consideration of s2w communications and can be analyzed using Theorem 1.6.
>
> **We hope that we have addressed all the questions. If you have any others, please let us know.**
>
> ---
>
> [1]: Yurii Nesterov. Lectures on convex optimization
>
> [2]: Yair Carmon, John C Duchi, Oliver Hinder, and Aaron Sidford. Lower bounds for finding stationary points i. Mathematical Programming, 184(1):71–120, 2020.
>
> [3]: Kumar Kshitij Patel, Lingxiao Wang, Blake Woodworth, Brian Bullins, and Nati Srebro. Towards optimal communication complexity in distributed non-convex optimization. NeurIPS 2022

---

> ### Author Response · Authors · 2025-11-27
>
> Dear Reviewer 2Z8R,
>
> Thank you for your time and effort. We are kindly asking whether you have had time to review our responses. We hope that we have addressed the questions and clarified the weaknesses. If you think that any questions require further discussion, we will be happy to elaborate.
>
> Thank you!
> Authors

---

> > ### Comment · Reviewer_2Z8R · 2025-11-28
> >
> > Thank you for addressing my concerns.
> >
> > Can you, please, elaborate more on Theorems 4.2 and F.1? Theorem 4.2 is stated with $\tau_s > 0$, but with $\tau_w = 0$. On the contrary, in Theorem F.1 $\tau_w > 0$, but $\tau_s = 0$. Please, correct me, if i am being wrong. How do we combine these statements and obtain the final bound, if they are proven under different assumptions?

---

> > > ### Author Response · Authors · 2025-11-28
> > >
> > > Thank you for the question.
> > >
> > > > Can you, please, elaborate more on Theorems 4.2 and F.1? Theorem 4.2 is stated with $\tau_s > 0$, but with $\tau_w = 0$. On the contrary, in Theorem F.1 $\tau_w > 0$, but $\tau_s = 0$. Please, correct me, if i am being wrong. How do we combine these statements and obtain the final bound, if they are proven under different assumptions?
> > >
> > > Yes, the previous version of the paper stated that Theorem 4.2 holds with $\tau_s > 0$ and $\tau_w = 0.$ However, the fact that the lower bound in Theorem 4.2 holds not only for $\tau_w = 0$ but for any $\tau_w \geq 0$ seemed to be a simple corollary, because if we start taking into account the communication time $\tau_w > 0,$ the total lower bound can only increase, not decrease. The same idea applies to Theorem F.1.
> > >
> > > To avoid any confusion, we improved the statements of Theorems 4.2 and F.1, and now they **both** support any $\tau_s > 0$ and $\tau_w > 0.$ **Please consider the updated PDF, where we've improved the statements.** Almost nothing changes in the proofs. We added Remarks E.2 (Line 1308) and F.2 (Line 1739), where we clarify why we can ignore the communication costs in the proofs.
> > >
> > > Thus, our results are correct, and there are no contradictions. We can combine both lower bounds into one and obtain Theorem 1.6.
> > >
> > > ---
> > >
> > > Thank you for your time! We hope that we have addressed all the concerns. If you have more questions, please let us know.

---

### Author Response · Authors · 2025-12-01
**Public Summary Comment by Authors**

Dear AC and Reviewers,

Thank you for your time and effort! We would like to provide a brief summary of the reviews and the rebuttal. In total, we believe that the reviewers are positive about our work:

> **Reviewer 2Z8R**: "Addressing not only w2s, but also s2w communications is important and not usually done in optimization." and "Modifying the worst-case function and using novel techniques to prove the lower bounds."

> **Reviewer 9bv8**: "**I think this paper may be accepted.**"

> **Reviewer vzMo**: "Overall, besides minor writing issues, I don't have any significant concerns with the paper and **recommend accepting it**. Once the authors address the writing issues, I am happy to increase my score."

> **Reviewer Neht**: "The explicit consideration of bidirectional communication costs (τs and τw) is a powerful and novel contribution, reflecting a more realistic model of distributed systems. 2.**The paper makes a significant theoretical contribution by constructing a new lower bound that clarifies the scalability limits of a specific class of optimization algorithms**. 3.**The construction of a novel "worst-case" function and the associated proof framework is a strong technical achievement**."

Reviewers 9bv8, vzMo, and Neht agree that our work should be accepted to the venue, and Reviewer vzMo even indicated (quote: “Once the authors address the writing issues, I am happy to increase my score.”) that the score would be increased after our clarifications and improvements (which, unfortunately, can not be done due to the leak incident). We want to emphasize that all questions and comments have been addressed and are already incorporated in the updated PDF, and, hopefully, Reviewer vzMo would increase the score to at least 8.

---

Reviewer 2Z8R gave us "Rating: 4" with "Confidence: 2" with the comments:

> The class of zero-respecting protocols is quite restricting, as it does not contain sketchings, dithered quantizations, and so on.

> Theorems 4.2 and E.1 seems to be proven only for RandK compressors, rather than for any unbiased zero-respecting.

In the rebuttal, we explain that the class of zero-respecting protocols is a standard class of methods considered in the community [1,2,3] and not restrictive at all. This family includes SGD-like, Adam-like, asynchronous SGD-like methods, and many other methods. Moreover,  this is standard practice for taking the “worst-case” compressor from the family (similarly to taking the “worst-case” functions (Carmon et al., 2020; Nesterov, 2018)). Thus, the fact that we focus only on the worst-case compressor is the standard practice. Even for RandK compressors and sparsifiers, our results are non-trivial, and extending them further is important future work.

Through the discussion, we also address the question

> Theorem 4.2 contains only $\tau_s$, but no $\tau_w$. Theorem E.1, on the contrary, contains $\tau_w$, but no $\tau_s$. Could the authors clarify, how they correctly combine lower bound, that involve both s2w and w2s communications? [...] Can you, please, elaborate more on Theorems 4.2 and F.1? Theorem 4.2 is stated with $\tau_s > 0$, but with $\tau_w = 0$. On the contrary, in Theorem F.1 $\tau_w > 0$, but $\tau_s = 0$. Please, correct me, if i am being wrong. How do we combine these statements and obtain the final bound, if they are proven under different assumptions?

In the rebuttal, we explain the fact that the lower bound in Theorem 4.2 holds not only for $\tau_w = 0$ but for any $\tau_w \geq 0$ seemed to be a simple corollary, because if we start taking into account the communication time $\tau_w > 0,$ the total lower bound can only increase, not decrease. The same idea applies to Theorem F.1. To avoid any confusion, we improved the statements of Theorems 4.2 and F.1, and now they both support any $\tau_s > 0$ and $\tau_w > 0.$ Please consider the updated PDF, where we've improved the statements. Almost nothing changes in the proofs. We added Remarks E.2 (Line 1308) and F.2 (Line 1739), where we clarify why we can ignore the communication costs in the proofs. Thus, our results are correct, and there are no contradictions. We can combine both lower bounds into one and obtain Theorem 1.6.

---

Thank you once again! We believe that the reviewers are positive about our work. All the weaknesses are minor and have already been fixed in the updated PDF.

---

[1]: Yurii Nesterov. Lectures on convex optimization

[2]: Yair Carmon, John C Duchi, Oliver Hinder, and Aaron Sidford. Lower bounds for finding stationary points i. Mathematical Programming, 184(1):71–120, 2020.

[3]: Kumar Kshitij Patel, Lingxiao Wang, Blake Woodworth, Brian Bullins, and Nati Srebro. Towards optimal communication complexity in distributed non-convex optimization. NeurIPS 2022

---

### Meta-Review · Area_Chair_V9tn · 2026-01-06

**Summary:**

The paper studies centralized distributed optimization under a time model that includes both worker-to-server and server-to-worker communication costs. Its main contribution is a new lower bound construction for the homogeneous nonconvex setting, built by modifying the standard chain hard instance to require a short block structure before new coordinates can influence progress. Under a random sparsification model (uniform random coordinate selection without replacement), the results show that once server-to-worker broadcast time is included, one cannot obtain meaningful scaling in both the noise-related term and the broadcast-related term beyond polylogarithmic factors. The paper also argues that, in common symmetric-cost regimes, the optimal time is essentially achieved by simple baselines such as minibatch SGD or single-worker SGD.

The central technical idea is novel, and the model choice is timely, but the scope is narrower than some of the broader framing suggests. The lower bound is established for zero-respecting protocols and for a specific family of sparsifiers; therefore, the practical claims should be stated with greater care. In particular, the discussion of biased compressors remains somewhat imprecise. However, the reviewers generally lean towards acceptance.

**Reviewer Concerns:**

Reviewer 2Z8R questions the restriction to zero-respecting protocols and the fact that key theorems are proved for RandK-style compressors rather than general unbiased compression. Reviewer vzMo asks for stronger related-work positioning and more tempered conclusions. The discussion clarifies how the separate lower bounds are combined and appears to fix a presentation issue that confused reviewer 2Z8R. The most important remaining issue is the conclusion-level interpretation, especially around biased compressors, which should be rewritten with clear definitions and careful quantifiers.

**Reviewer Scores:**

Reviewer 9bv8 and Neht likely remain with their score of 6, as their remaining concerns are not fully resolved.
Reviewer 2Z8R possibly migth increase from 4 to 6, as the rebuttal seems to provide clarification on combining the bounds.
Reviewer vzMo could possibly be increased to 8 (or remain at 6), given the added related work discussion.

---

### Decision · Program_Chairs · 2026-01-26

Accept (Poster)